



# The surface energy balance in a cold-arid permafrost environment,

# Ladakh Himalaya, India

John Mohd Wani[1], Renoj J. Thayyen[2*], Chandra Shekhar Prasad Ojha[1], and Stephan Gruber[3]

[1]Department of Civil Engineering, Indian Institute of Technology (IIT) Roorkee, India, [2]Water Resources System Division, National Institute of Hydrology, Roorkee, India (renoj.nihr@gov.in; renojthayyen@gmail.com), [3]Department of Geography & Environmental Studies, Carleton University, Ottawa, Canada

**Abstract:** Cryosphere of the cold-arid trans-Himalayan region is unique with its significant permafrost cover. While the information on the permafrost characteristics and its extent started emerging, the governing energy regimes of this cryosphere region is of particular interest. This paper present the results of Surface Energy Balance (SEB) studies carried out in the upper

Ganglass catchment in the Ladakh region of India, which feed directly to the River Indus. The point SEB is estimated using the one-dimensional mode of GEOtop model from 1 September 2015 to 31 August 2017 at 4727 m a.s.l elevation. The model is evaluated using field monitored radiation components, snow depth variations and one-year near-surface ground temperatures and showed good agreement with the respective simulated values. The study site has an air

temperature range of – 23.7 to 18.1 °C with a mean annual average temperature (MAAT) of - 2.5 and ground surface temperature range of  -9.8 to 19.1 °C. For the study period, the surface energy balance characteristics of the cold-arid site show that the net radiation was the major component with mean value of 28.9 W m$^{-2}$, followed by sensible heat flux (13.5 W m$^{-2}$) and latent heat flux (12.8 W m$^{-2}$), and the ground heat flux was equal to 0.4 W m$^{-2}$. The partitioning

of energy balance during the study period shows that 47% of $R_n$ was converted into H, 44% into LE, 1% into G and 7% for melting of seasonal snow. Both the study years experienced distinctly different, low and high snow regime. Key differences due to this snow regime change in surface energy balance characteristics were observed during peak summer (July-August).





The latent heat flux was higher (lower) during this period with 39 W m$^{-2}$ (11 W m$^{-2}$) during
high (low) snow years. The study also shows that the sensible heat flux during the early summer
season (May, June) of the high (low) snow was much smaller (higher) -3.4 W m$^{-2}$ (36.1 W m$^{-2}$). During the study period, snow cover builds up in the catchment initiated by the last week of
December facilitating the ground cooling by almost three months (October to December) of
sub-zero temperatures up to -20 °C providing a favourable environment for permafrost. It is
observed that the Ladakh region have a very low relative humidity in the range of 43% as
compared to, e.g., ~70% in the Alps facilitating lower incoming longwave radiation and
strongly negative net longwave radiation averaging ~ -90 W m$^{-2}$ compared to -40 W m$^{-2}$ in the
Alps. Hence, the high elevation cold-arid region land surfaces could be overall colder than the
locations with more RH such as the Alps. Further, it is apprehended that high incoming
shortwave radiation in the region may be facilitating enhanced cooling of wet valley bottom
surfaces as a result of stronger evaporation.

**Keywords:** Cold-arid, Cryosphere, GEOtop, Himalaya, Leh, Surface Energy Balance,
Permafrost

## 1 Introduction

The Himalayan cryosphere is important for sustaining the flows in the major rivers originating
from the region (Bolch et al., 2012; Immerzeel et al., 2012; Kaser et al., 2010; Lutz et al., 2014;
Pritchard, 2019). These rivers flow through the most populated regions of the world and insight
on the processes driving the change is critical for evaluating the future trajectory of water
resources of the region, ranging from small headwater catchments to large river systems. It is
hard to propose a uniform framework for the downstream response of these rivers as they
originate and flow through various glacio-hydrological regimes of the Himalaya (Thayyen and
Gergan, 2010). Lack of understanding of various processes driving the cryospheric response of
the region is limiting our ability to anticipate the ensuing changes and their impacts correctly.



This has been highlighted by the recent studies which suggested the occurrence of higher

precipitation in the accumulation zones of the glaciers than previously known (Immerzeel et

al., 2015).

The role of permafrost is another key unknown variable in the Himalaya, especially in the Indus

basin. Recent studies have signalled significant permafrost cover in the cold-arid upper Indus

basin areas covering Ladakh (Wani et al., 2020). This study suggests permafrost cover in a

small (15.4 km$^2$ ) catchment in the Ladakh region is 22 times of the glacier area. More coarse

assessment in the Hindu Kush Himalaya (HKH) region suggests that the permafrost area

extends up to 1 million km$^2$, which roughly translate into 14 times the area of glacier cover of

the region (Gruber et al., 2017). Except for Bhutan, the expected permafrost areas in all other

countries is larger than the glacier area. With two-thirds of the HKH underlain by permafrost,

China has by far the largest share (906x10$^3$ km$^2$) followed by India (40.1x10$^3$ km$^2$), Pakistan

(26.6x10$^3$ km$^2$), Afghanistan (17.5x10$^3$ km$^2$), Nepal (11.1x10$^3$ km$^2$), Bhutan (1.2x10$^3$ km$^2$) and

Myanmar (0.1x10$^3$ km$^2$) (cf. Table 1, Gruber et al., 2017). The mapping of rock glaciers using

remote sensing suggested that the discontinuous permafrost in the HKH region can be found

between 3500 m a.s.l. in Northern Afghanistan to 5500 m a.s.l. on the Tibetan Plateau (Schmid

et al., 2015). Recently, Pandey (2019) published the first remote sensing based rock glacier

inventory of Himachal Himalaya, which falls in the Indian Himalayan Region (IHR). The

inventory reports that the discontinuous permafrost can be found within an elevation range of

3052–5503 m a.s.l. Another rock glacier inventory for Uttarakhand State, India suggests that

the higher elevation regions above  4600 m a.s.l. are suitable for the occurrence of permafrost

(Baral et al., 2019). Similarly, an initial localised estimate of 420 km$^2$ of permafrost is

suggested in the Kullu district of Himachal Pradesh, India (Allen et al., 2016).

The cold-arid region of Ladakh has reported sporadic occurrence of permafrost and associated

landforms (Gruber et al., 2017; Wani et al., 2020) with the sorted patterned ground and other



periglacial landforms such as ice-cored moraines. Previous studies of permafrost in the Ladakh

region are from the Tso Kar basin (Rastogi and Narayan, 1999; Wünnemann et al., 2008), and

a recent one from the Changla region, where the depth of permafrost table was found to be

~110 cm (Ali et al., 2018). Field observations suggest that ground-ice melt may be a critical

water source in dry summer years in the cold-arid regions of Ladakh (Thayyen, 2015).

The energy balance at the earth's surface drives the Spatio-temporal variability of ground

temperature (Westermann et al., 2009). It is linked to the atmospheric boundary layer, and

location-dependent transfer mechanisms between land and the overlying atmosphere (Endrizzi,

2007; Martin and Lejeune, 1998; McBean and Miyake, 1972). The surface energy balance

(SEB) in cold regions additionally depends on the seasonal snow cover, vegetation and

moisture availability in the soil (Lunardini, 1981) and (semi-) arid areas exhibit their typical

characteristics (Xia, 2010). The SEB characteristics of different permafrost regions have been

studied, e.g., the North American Arctic (Eugster et al., 2000; Lynch et al., 1999; Ohmura,

1982, 1984), European Arctic (Lloyd et al., 2001; Westermann et al., 2009), Tibetan Plateau

(Gu et al., 2015; Yao et al., 2011), European Alps (Mittaz et al., 2000) or Siberia (Boike et al.,

2008; Kodama et al., 2007; Langer et al., 2011a, 2011b). However, SEB studies of IHR  is very

limited (Azam et al., 2014). This highlight that the knowledge of frozen ground and associated

energy regimes are a key knowledge gap in our understanding of the Himalayan cryospheric

systems, especially in the Upper Indus Basin.

This study presents a SEB analysis of a permafrost environment in the cold-arid trans-

Himalaya, where a recent study has identified significant permafrost cover (Wani et al., 2020).

With this, we aim to provide a foundation for better understanding the micro-climatological

drivers affecting permafrost distribution and temperature regimes in the area, to build

hypotheses about similarities and major differences with other, better-investigated permafrost

areas. This is important to guide the application of models calibrated (Boeckli et al., 2012) or



tested (Cao et al., 2019; Fiddes et al., 2015) elsewhere for further investigations in the Ladakh

region, where only little data on ground temperatures and permafrost are currently available. It

will also help to interpret differences in the relationships of air and shallow ground

temperatures (surface offset) observed in Ladakh (Wani et al., 2020) and other permafrost areas

(Boeckli et al., 2012; Hasler et al., 2015; PERMOS, 2019).

The specific objectives of this study are to (a) quantify the point Surface Energy Balance (SEB)

and its components in a cold-arid Himalayan permafrost environment, (b) evaluate the quality

of SEB assessment by modelling snow depth and near-surface ground temperature variations

and compare with the field observations (c) understand the role of winter snowpack

characteristics (timing, thickness and duration) and its effect on ground surface temperature,

and (d) compare the SEB regime of cold-arid Himalaya with other better-investigated

permafrost regions of the world.

## 2 Materials and methods

### 2.1 Study area

The present study is carried out at South-Pullu (34.25°N, 77.62°E, 4727 m a.s.l.) in the upper

Ganglass catchment (34.25°N to 34.30°N and 77.50°E to 77.65°E), Leh, Ladakh (Figure 1).

Ladakh is a Union territory of India and has a unique climate, hydrology and landforms. Leh

is the district headquarter, where long-term climate data is available (Bhutiyani et al., 2007).

Long-term mean precipitation of Leh (1908–2017, 3526 m a.s.l.) is 115 mm (Lone et al., 2019;

Thayyen et al., 2013) and the daily minimum and maximum temperatures during the period

(2010 to 2012) range between -23.4 to 33.8 °C (Thayyen and Dimri, 2014). The spatial area of

the catchment is 15.4 km$^2$ and extends from 4700 m to 5700 m a.s.l. A small cirque glacier

called as Phuche glacier with an area of 0.62 km$^2$ occupies the higher elevations of the

catchment. A single stream flows through the valley of the catchment originating from Phuche

glacier. This stream flows intermittently with most of the flow from May to October.
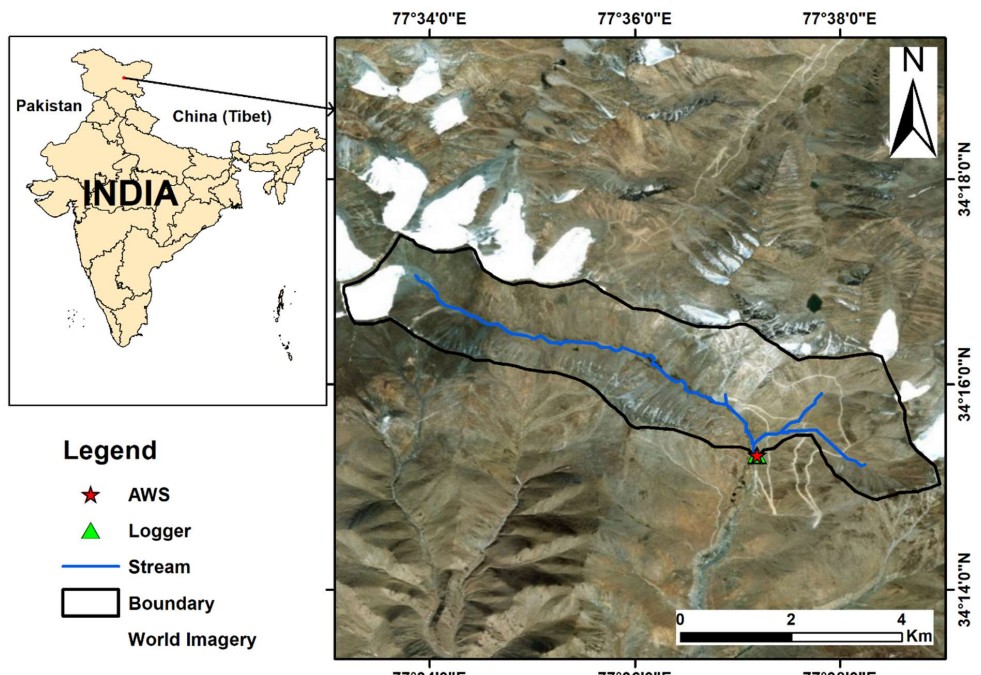

Figure 1 Location of the study site in the upper Ganglass catchment. (Base image layer sources on the right panel: © Esri, DigitalGlobe, GeoEye, Earthstar Geographic's, CNES/Airbus DS, USDA, USGS, AEX, Getmapping, Aerogrid, IGN, IGP, swisstopo, and the GIS User Community).

The catchment lies in the Ladakh mountain range and is part of the main Indus river basin.

Geologically, the study catchment is part of the Ladakh batholith. It mainly comprises of quartz-bearing rocks such as quartz diorite, quartz-monzodiorite, granodiorite, monzonite and granite with occasional masses of diorite, gabbro, pyroxenite and anorthosite (Priya et al., 2016; Trivedi et al., 1982). Based on the $SiO_2$ and total alkalis content of these rock types, they are classified as sub-alkaline granites (Upadhyay et al., 2008). The granites of the Ladakh batholite

are generally calc-alkaline and belong to type I (Jowhar, 2001; Sharma and Choubey, 1983). In the Leh-Khardung La section, hornblende-biotite granites and granodiorites are the dominant rock types (Jowhar, 2001). The edges of Ladakh batholith comprise of xenoliths of



basic (metavolcanics, diorite and amphibolite) and metamorphic (quartzite, mica schist and marble) rocks (Sharma and Choubey, 1983; Thakur, 1990).

### 2.2 Meteorological data used

The automatic weather station (AWS) in the catchment is located at an elevation of 4727 m a.s.l. at South-Pullu (Figure 1). It is located in the wide deglaciated valley trending southeast. The site has a local slope angle of 15°, and the soil is sparsely vegetated. Weather data has been collected by a Sutron automatic weather station from 1 September 2015 to 31 August 2017. The study years 1 September 2015 to 31 August 2016 and 1 September 2016 to 31 August 2017 hereafter in the text will be designated as 2015-16 and 2016-17 respectively. The variables measured include air temperature, relative humidity, wind speed and direction, incoming and outgoing shortwave and longwave radiation and snow depth (Table 1). The snow depth is measured using a Campbell SR50 sonic ranging sensor with a nominal accuracy of ±1 cm (Table 1). The analysis of data was performed using R (R Core Team, 2016; Wickham, 2016, 2017; Wickham and Francois, 2016; Wilke, 2019). To reduce the noise of the measured snow depth, a six-hour moving average is applied. Near-surface ground temperature (GST) is measured at a depth of 0.1 m near the AWS using miniature temperature data logger (MTD) manufactured by GeoPrecision GmbH, Germany. GST data was available only from 1 September 2016 to 31 August 2017 and is used for model evaluation, only. All the four solar radiation components, i.e., incoming shortwave ($SW_{in}$), outgoing shortwave ($SW_{out}$), incoming longwave ($LW_{in}$) and outgoing longwave ($LW_{out}$) radiation were measured. Before using these data in the SEB calculations, necessary corrections were applied (Nicholson et al., 2013; Oerlemans and Klok, 2002): (a) all the values of $SW_{in} < 5$ $Wm^{-2}$ are set to zero, (b) when $SW_{out} > SW_{in}$ (3 % of data understudy), it indicates that the upward-looking sensor was covered with snow (Oerlemans and Klok, 2002). The $SW_{out}$ can be higher than $SW_{in}$ at high elevation sites such as this one due to high solar zenith angle during the morning and evening hours (Nicholson



et al., 2013). In such cases, $SW_{in}$ was corrected by $SW_{out}$ divided by the accumulated albedo,

calculated by the ratio of measured $SW_{out}$ and measured $SW_{in}$ for a 24h period (van den Broeke

et al., 2004).

Table 1 Technical parameters of different sensors at South-Pullu (4727 m a.s.l.) in the upper

Ganglass catchment, Leh. (MF: model forcing, ME: model evaluation).

| Variable | Units | Sensor | Stated accuracy | Height (m) | Use |
|---|---|---|---|---|---|
| Air temperature | (°C) | Rotronics-5600-0316-1 | ±0.2 °C | 2.2 | MF |
| Relative humidity | (%) | Rotronics-5600-0316-1 | ±1.5% | 2.2 | MF |
| Wind speed | (m s$^{-1}$) | RM Young 05103-45 | ±0.3 ms$^{-1}$ | 10 | MF |
| Wind direction | (°) | RM Young 05103-45 | ±0.3° | 10 | MF |
| In/out shortwave radiation | (W m$^{-2}$) | Kipp and Zonen (CMP6) (285 to 2800nm) | ±10% | 4.6 | MF/ME |
| In/out longwave radiation | (W m$^{-2}$) | Kipp and Zonen (CGR3) (4500 to 42000nm) | ±10% | 4.3 | ME |
| Snow depth | (m) | Campbell SR-50 | ±1cm | 3.44 | ME |
| Data platform | - | Sutron 9210-0000-2B | - | - | - |
| Near-surface ground temperature | (°C) | PT1000 in stainless steel cap (by GEOPRECISION GmbH, Germany) | ±0.1 °C | -0.1 | ME |

### 2.3    Estimation of precipitation from snow height (ESOLIP approach)

In high elevation and remote sites, the snowfall measurement is a difficult task with an under

catch of 20–50% (Rasmussen et al., 2012). At the South Pullu station, daily precipitation

including snow was measured using a non-recording rain gauge. It is a known fact that the

snow water equivalent measurements in the mountainous region using collectors have

significant errors due to under catch (Yang et al., 1999). In this high elevation area, an under

catch of 23% of snowfall was reported earlier (Thayyen et al., 2015) [Unpublished work]. To

improve the data quality and match the temporal resolution of precipitation data with other

meteorological forcing's, we adopted the method proposed by Mair et al. (2016), called

Estimating SOlid and Liquid Precipitation (ESOLIP). This method makes use of snow depth





and meteorological observations to estimate the winter precipitation. In this method, the sub-

daily solid precipitation is estimated in terms of snow water equivalent (SWE). To estimate the

SWE of single snowfall events from snow depth measurements, an identification of the snow

height increments of the single snowfall events and an accurate estimate of the snow density

are necessary. The fresh snow density was estimated based on air temperature and wind speed

as below (Jordan et al., 1999):

$$\rho = 500 * [1 - 0.951 * exp(-1.4 * (278.15 - T_a)^{-1.15} - 0.008 U_{10}^{1.7})], \quad (1)$$


For $260.15 < Ta \leq 275.65$ K

$$\rho = 500 * [1 - 0.904 * \exp(-0.008 U_{10}^{1.7})], \quad (2)$$

For $Ta \leq 260.15$ K

## 2.4     Modelling point surface energy balance

In this study, the open-source GEOtop 2.0 version (hereafter GEOtop) (Endrizzi et al., 2014)

was used for the modelling of point surface energy balance. GEOtop is a physically-based fully

distributed model for modelling of water and energy balances at and below the soil surface. It

represents the combined ground heat and water balance, the exchange of energy with the

atmosphere by taking into consideration the radiative and turbulent heat fluxes. The model also

has a multi-layer snowpack and solves the energy and water balance of the snow cover.

Furthermore, the temporal evolution of snow depth and its effect on soil temperature are

simulated. The GEOtop also simulates the highly non-linear interactions between the water and

energy balance during soil freezing and thawing (Dall'Amico et al., 2011; Endrizzi et al.,

2014). The model solves Richard's equation in three or one dimensions, and the heat equation

in one dimension (1D) (Endrizzi et al., 2014). It can be applied in high mountain regions with

complex terrain and makes it possible to account for topographical and other environmental





variability (Fiddes et al., 2015; Gubler et al., 2013). The model takes into account the effects

of complex topography in the estimation of radiation components (Endrizzi et al., 2014), such

as (i) the incoming solar radiation is partitioned into direct and diffuse components according

to Erbs et al. (1982), (ii) taking into account the solar incidence angle and shadowing of direct

incoming solar radiation by topography, (iii) the effects of topography on diffuse radiation

coming on the surrounding terrain (Iqbal, 1983).

The model can be operated in two configurations, either in pointwise (1D) or distributed mode

(2D) and the processes of interest can be controlled through parameters (Endrizzi et al., 2014).

Previous studies have successfully applied GEOtop in mountains regions, e.g., simulating snow

depth and ground temperature (Endrizzi et al., 2014), snow cover mapping (Dall'Amico et al.,

2018; Dall'Amico et al., 2011; Zanotti et al., 2004), ecohydrological processes (Bertoldi et al.,

2010), modelling of processes in complex topography (Fiddes and Gruber, 2012), permafrost

distribution (Fiddes et al., 2015) or modelling ground temperatures (Gubler et al., 2013). In this

study, only the energy fluxes over the snow cover and the ground surface in one-dimensional

(1D) mode of GEOtop are used.

Generally, the surface energy balance (SEB) (Eq. 3) is written as a combination of net radiation

($R_n$), sensible (H) and latent heat (LE) flux and heat conduction into the ground or to the snow

(G) and must balance at all times (Oke, 2002; Williams and Smith, 1989):


$$R_n - H - LE - G = F_{surf} \qquad\qquad (3)$$

where $F_{surf}$ is the resulting energy flux at the surface. During the summertime when the melting

conditions prevail, the $F_{surf}$ is positive and is the energy available for melting snow; otherwise,

$F_{surf}$ is equal to zero.

But in the GEOtop (Endrizzi et al., 2014) the equations of SEB are described separately. The

equations and the key elements of GEOtop are explained in Endrizzi et al. (2014), and here,





only a brief description of the equations that are of interest in this study is given. In GEOtop, the surface heat flux ($Q_s$) is the energy available for exchange and is given by the sum of net shortwave ($SW_n$) and net longwave ($LW_n$) radiations and turbulent heat fluxes i.e. sensible (H)

and latent heat flux (LE). The surface heat flux equation (Eq. 4) used in GEOtop is given below:

$$Q_s(T_s) = SW_n + LW_n(T_s) - H(T_s) - LE(T_s, \theta_w) \qquad (4)$$

where $T_s$, the temperature of the surface, is an unknown in the equation. The $Q_s$ is a function of the temperature at the surface ($T_s$), which is an unknown in the equation. Other terms in Eq.

4 which are a function of $T_s$ include $LW_n$, H and LE. In addition, the LE also depends on the soil moisture at the surface ($\theta_w$), linking the SEB and water balance equations. In Eq. 4, the sum of $SW_n$ and $LW_n$ is equal to the net radiation ($R_n$) (Oke, 2002). The sign convention adopted is as energy is considered as a gain for the surface or system if $R_n$ is positive and negative for H and LE. Conversely, energy is considered as loss for the surface or system, if

$R_n$ is negative and positive for H and LE (Endrizzi, 2007; Oke, 2002).

The $SW_n$ term in Eq. 4 is equal to the difference between the incoming solar radiation ($SW_{in}$) coming from the atmosphere and the reflected shortwave radiation ($SW_{out}$) (Oke, 2002). The $SW_{out}$ is given by $SW_{in}$ multiplied by the broadband albedo. The $SW_{in}$ (Eq. 5) on a flat ground surface is the product of shortwave radiation at the top of the atmosphere ($SW_{toa}$), atmospheric

transmissivity ($\tau_a$) and cloud transmissivity ($\tau_c$) as follows:

$$SW_{in} = SW_{toa}.\tau_a.\tau_c \qquad (5)$$

The albedo in GEOtop (Endrizzi et al., 2014) is considered as per the ground surface conditions such as, for the snow-free ground, the albedo varies linearly with the water content of the





topsoil layer, and for snow-covered surfaces, the albedo is estimated according to the Biosphere

Atmosphere Transfer Scheme (BATS) (Dickinson et al., 1993).

Also, $LW_n$ in Eq. 4 is equal to the difference between the incoming longwave radiation ($LW_{in}$)

coming from the atmosphere and the outgoing longwave radiation ($LW_{out}$) radiated by the

surface (Oke, 2002). $LW_{in}$ received at the surface is the combined result of the radiations

radiated at different heights in the atmosphere with different temperatures and gas

concentrations. The clear sky incoming longwave radiation ($LW_{in,clear}$) is calculated with one

of the nine parameterizations present in the model (see Endrizzi et al., 2014). Generally, these

parameterisations apply the Stefan-Boltzmann law (Eq. 6), which depends on near-surface air

temperature ($T_a$) in Kelvin, effective atmosphere emissivity $\epsilon_a$ (-) a function of $T_a$ (K) and

water vapour pressure $e_a$ (bar), as below:

$$LW_{in,clear} = \epsilon_a \ (T_a, e_a). \sigma. T_a^4 \qquad (6)$$

where $\sigma$ is the Stefan-Boltzmann constant equal to $5.67 \times 10^8$ Wm$^{-2}$ K$^{-4}$. The parameterisations

in GEOtop mainly differ in the value of effective atmosphere emissivity ($\epsilon_a$). In GEOtop

(Endrizzi et al., 2014), the cloudy conditions are treated according to the formulation of

(Crawford and Duchon, 1999) as below (Eq. 7):

$$\epsilon_c = \tau_c + (1 - \tau_c). \epsilon_a \qquad (7)$$

where $\epsilon_c$ is the cloudy sky atmosphere emissivity and $\tau_c$ is the shortwave radiation cloud

transmissivity. The $LW_{out}$ radiated by the surface is also estimated using the Stefan-Boltzmann

law (Eq. 8), as below:

$$LW_{out} = \epsilon_s. \sigma. T_s^4 \qquad (8)$$



where $T_s$ is the surface temperature (K) and $\in_s$ is the surface emissivity.

The turbulent fluxes (H and LE) are driven by the gradients of temperature and specific humidity between the air and the surface, and due to turbulence caused by winds as main transfer mechanism in the boundary layer (Endrizzi, 2007). GEOtop estimates the turbulent heat fluxes H (Eq. 9) and LE (Eq. 10) using the flux-gradient relationship (Brutsaert, 1975; Garratt, 1994) as below:

$$H = \rho_a c_p w_s \frac{T_a - T_s}{r_a} \qquad (9)$$


$$LE = \beta_{YP} L_e \rho_a c_p w_s \frac{Q_a - \alpha_{YP} Q_s^*}{r_a} \qquad (10)$$

where $\rho_a$ is the air density (kg m$^{-3}$), $w_s$ is the wind speed (m s$^{-1}$), $c_p$ the specific heat at constant pressure (J kg$^{-1}$ K$^{-1}$), $L_e$ the specific heat of vaporisation (J kg$^{-1}$), $Q_a$ and $Q_s^*$ are the specific humidity of the air (kg kg$^{-1}$) and saturated specific humidity at the surface (kg kg$^{-1}$)

respectively, and $r_a$ is the aerodynamic resistance (-). The $\beta_{YP}$ and $\alpha_{YP}$ are the coefficients that take into account the soil resistance to evaporation, and only depend on the liquid water pressure close to the soil surface. They are calculated according to the parameterisation of Ye and Pielke (1993), which considers evaporation as the sum of the proper evaporation from the surface and diffusion of water vapour in soil pores at greater depths. The aerodynamical

resistance is obtained applying the Monin–Obukhov similarity theory (Monin and Obukhov, 1954), which requires that values of wind speed, air temperature and specific humidity are available at least at two different heights above the surface.

The input meteorological data required for running the 1D GEOtop model include time series of precipitation, air temperature, relative humidity, wind speed, wind direction and solar

radiation components and the description of the topography (slope angle, elevation, aspect angle, and sky view factor) for the simulation point. Also, the latitude and longitude of the





study area have to be defined to allow the model to calculate the solar zenith angle, which is

important for shadowing estimations.

### 2.4.1 Heat equation

The equation (Endrizzi et al., 2014) representing the energy balance in a soil volume subject to

phase change is given below (Eq. 11):

$$\frac{\partial U^{\mathrm{ph}}}{\partial t} + \nabla \cdot \mathbf{G} + S_{en} - \rho_w[L_f + c_w(T - T_{ref})]S_w = 0 \tag{11}$$

where $U^{\mathrm{ph}}$ is the volumetric internal energy of soil (J m$^{-3}$) subject to phase change, t(s) time,

$\nabla\cdot$ the divergence operator, G the heat conduction flux (W m$^{-2}$), $S_{en}$ is the energy sink term

(W m$^{-3}$), $S_w$ is the mass sink term (s$^{-1}$), $L_f$ (J kg$^{-1}$) the latent heat of fusion, $\rho_w$ the density of

liquid water in soil (kg m$^{-3}$), $c_w$ is the specific thermal capacity of water (J kg$^{-1}$ K$^{-1}$), T (°C)

the soil temperature and $T_{ref}$ (°C) the reference temperature at which the internal energy is

calculated. The detailed description of the heat conduction equation used in GEOtop can be

found in Endrizzi et al. (2014).

### 2.4.2 Modelling of snow depth

The snow cover buffers the energy exchange between the soil and atmosphere and critically

influences the soil thermal regime. In GEOtop, the equations for snow modelling are similar to

the ones used for the soil matrix (Endrizzi et al., 2014). The snow processes are solved in a

particular order such as (i) solving the heat equation for snow, (ii) metamorphism of the

snowpack, (iii) water percolation in the snow and (iv) accumulation due to snow precipitation

(Endrizzi et al., 2014).

### 2.4.3 Model setup and forcing

The 1D GEOtop simulation was carried out at South-Pullu (Figure 1). The soil column is

discretised into 19 layers, with thickness increasing from the surface to the deeper layers. The

top 8 layers close to the ground surface were resolved with thicknesses ranging from 0.1 to 1



m, because of the higher temperature and water pressure gradients near the surface (Endrizzi et al., 2014), while the lowest layer is 4.0 m thick.

The snowpack is discretised in 10 layers, which are finer at the top with the atmosphere and at the bottom with the soil, than in the middle. At the top and bottom regions of the snowpack,

the vertical gradients are high because of the interactions with the atmosphere at top and the soil at the bottom (Endrizzi et al., 2014).

The model was initialised at a uniform temperature of -0.5 ºC. The soil column in the model is 10 m deep. The model is initialised by repeatedly modelling the soil temperature down to 1 m (2 years*25 times = 50 years), then using the modelled soil temperatures as an initial condition

to repeatedly simulate soil temperature down to 10 m (50 years) and finally simulating soil temperatures down to 10 m depth. This initialisation technique has been successfully applied in earlier work (Fiddes et al., 2015; Gubler et al., 2013; Pogliotti, 2011). Preliminary tests show that the minimum number of repetitions required to bring the soil column to equilibrium was 25 (Figure *S1*).

In this study, the data recorded by the AWS was used as model forcing, and the forcing data consist of hourly air temperature, wind speed and direction, and global incoming shortwave radiation. The ESOLIP estimated hourly precipitation was also used as forcing to the model. The model runs at an hourly time step corresponding to the measurement time step of the meteorological data.

**2.5    Model performance evaluation**

The evaluation of point SEB was done based on three variables such as radiation components, snow depth and the GST. These variables were chosen because they represent different physical processes influencing the surface energy balance at the ground. For example, (a) the radiation components are the main input driving the surface energy balance, (b) the melt-out

date of the snow depth is a good indicator showing how good the surface energy balance is





simulated and (c) the GST is the result of all the processes occurring at the ground surface such

as radiation, turbulence, latent and sensible heat fluxes (Gubler, 2013). Model performance is

evaluated based on the measured and the simulated time series (Gubler et al., 2012). Typically,

a variety of statistical measures are used to evaluate the model performance because no single

measure enclose all aspects of interest. In this study also, different model evaluation statistics

were used for (a) radiation components, and (b) GST and the snow depth as described below.

### 2.5.1 Performance statistics for evaluation of radiation components

For the evaluation of radiation components, we prefer the statistics mean bias difference

(MBD) and the root mean square difference (RMSD) (Badescu et al., 2012; Gubler et al., 2012;

Gueymard, 2012). These statistics indicate model prediction accuracy (Stow et al., 2003). The

MBD (Eq. 12) is a simple and familiar measure that neglects the magnitude of the errors (i.e.

positive errors can compensate for negative ones) (Gubler et al., 2012):

$$MBD = \frac{1}{\overline{y^*}} \frac{\sum_{t=1}^{n}(y_t - y_t^*)}{n} \qquad (12)$$

Here, $y_t$ is the modelled output variable, and $y_t^*$ is the corresponding measured variable. The

MBD ranges from $-\infty$ to $\infty$. The perfect model is the one with an MBD value equal to 0. The

RMSD (Eq. 13) is calculated as (Gubler et al., 2012):

$$RMSD = \frac{1}{\overline{y^*}} \sqrt{\frac{1}{n} \sum_{t=1}^{n}(y_t - y_t^*)^2} \qquad (13)$$

The RMSD takes into account the average magnitude of the errors and puts weight on larger

errors, but neglects the direction of the errors (Gubler et al., 2012). The RMSD ranges from 0

to $\infty$. The perfect model is the one with an RMSD value equal to 0. The formulae (Eq. 12 and

13) used for estimation of MBD and RMSD respectively provide dimensionless quantities





since their right-hand side has been divided by the mean of the measured variable (Badescu et al., 2012). Hence, they are expressed in per cent throughout the manuscript for clarity. Also,

we use the coefficient of determination ($R^2$, Eq. 14) defined as the squared value of the coefficient of correlation which indicates the amount of variation in the modelled variable predictable from the measured variable:

$$R^2 = \left( \frac{\sum_{t=1}^{n}(y_t - \bar{y})\,(y_t^* - \overline{y^*})}{\sqrt{\sum_{t=1}^{n}(y_t - \bar{y})^2(y_t^* - \overline{y^*})^2}} \right)^2 \tag{14}$$

### 2.5.2 Performance statistics for evaluation of snow depth and near-surface ground

temperature

For the evaluation of near-surface ground temperature and snow depth apart from the coefficient of determination ($R^2$), different statistical measures have been used such as mean bias (MB), and root means square error (RMSE).

The MB (Eq. 15) provides a good indication of the mean over or underestimate of predictions

(Carslaw and Ropkins, 2012). MB is in the same units as the variables being considered.

$$MB = \frac{1}{n}\sum_{i=1}^{n}(y_t - y_t^*) \tag{15}$$

The optimal value of MB is equal to zero. The positive and negative MB values indicate model over-estimation and under-estimation bias, respectively.

The RMSE (Moriasi et al., 2007) is a commonly used statistic that provides a good overall measure of how close modelled values are to predicted values and is given below (Eq. 16):

$$RMSE = \sqrt{\frac{1}{n}\sum_{t=1}^{n}(y_t - y_t^*)^2} \tag{16}$$


# 3    Results

## 3.1    Meteorological characteristics

A summary of the meteorological conditions measured at South-Pullu (4727 m a.s.l.) study site

is given in Table 2 to provide an overview of the prevailing weather in the study region. The

daily mean air temperature ($T_a$) throughout the study period varies between -19.5 to 13.1 °C

with a mean annual average temperature (MAAT) of -2.5 °C (Figure 2A). The $T_a$ shows

significant seasonal variations and instantaneous hourly temperature at the study site range

between -23.7 °C in January and 18.1 °C in July. During the two year study period, sub-zero

mean monthly temperature prevailed for seven months from October to April in both the years

(2015-16 and 2016-17). The monthly mean $T_a$ during pre-winter months (September to

December) of 2015-16 and 2016-17 was −4.6 and -2.7 °C respectively. During the core winter

months (January to February) of 2015-16 and 2016-17, the respective monthly mean $T_a$ was -

13.1 and -13.7 °C,  for post-winter months (March and April), mean monthly $T_a$ was -5.8 and

-8 °C, respectively and for summer months (May to August), the respective monthly mean $T_a$

was 6.6 and 5.5 °C. A sudden change in the mean monthly $T_a$ characterises the onset of a new

season, and the most evident inter-season change was found between the winter and summer

with a difference of about 16 °C during both the years.

The mean daily GST recorded by the logger near the AWS available for one year (1 September

2016 to 31 August 2017) is also plotted along with air temperature (Figure 2A). The mean daily

GST ranges from -9.1 to 15.1 °C with mean annual GST of 2.1 °C. The instantaneous hourly

GST at the study site range between -9.8 °C in December and 19.1 °C in July. The GST

followed the pattern of air temperature, but during winter, the snow cover dampened the

pattern. The GST was higher than the $T_a$ except for a short period during snowmelt.

Mean relative humidity (RH) was equal to 43% during the study period (Figure 2B). The

instantaneous hourly values of RH at the study site range between 3% (1 October 2016) and



100% (22 September 2015). The daily mean RH greater than 50% and 80% was recorded on

224 and 18 days respectively. The average RH during the pre-winter months (September to

December) of the 2015-16 year was greater (39%) in comparison to RH (28%) recorded during

pre-winter months of the 2016-17 year. However, during core winter months (January to

February) of the 2015-16 year was smaller (42%) in comparison to the RH (53%) recorded

during core winter months of the 2016-17 year. The average RH during the summer months

(May to August) of the 2015-16 year was smaller (45%) in comparison to RH (51%) recorded

during summer months of the 2016-17 year. Furthermore, for the low snow year (2015-16), the

annual mean RH was 42.7%, and for the high snow year (2016-17) the annual mean RH was

44%.

The daily average wind speed ($u$) ranges between 0.6 (29 January 2017) to 7.1 m s$^{-1}$ (6 April

2017) with a mean wind speed of 3.1 m s$^{-1}$ (Figure 2C). The hourly maximum instantaneous

value of $u$ recorded was 11.1 m s$^{-1}$ (4 February 2017). The annual average $u$ was 3.2 and 2.9 m

s$^{-1}$ during the years 2015-16 and 2016-17 respectively. The instantaneous hourly $u$ was plotted

as a function of wind direction (WD) (Figure *S2*) for the study period which shows that there

is a persistent dominance of katabatic and anabatic winds at the study site, which is typical of

a mountain environment. The average WD during the study period was southeast (148°)

(Figure 2D).





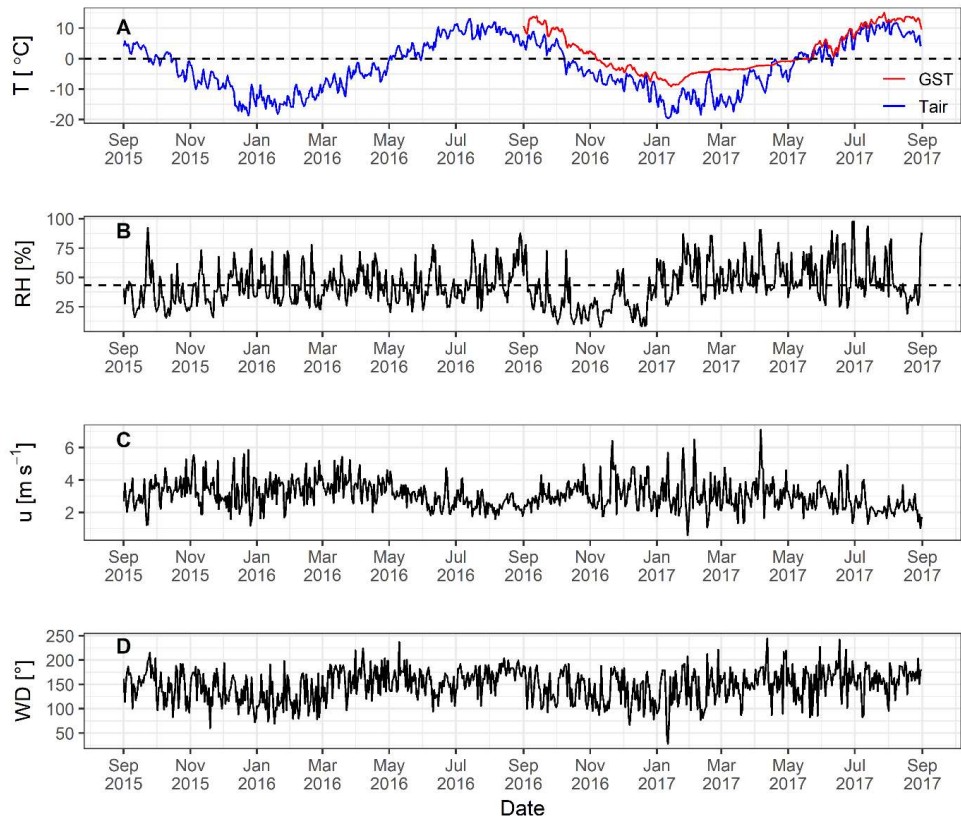

Figure 2 Daily mean values of observed (A) air temperature (Ta, blue) and one-year GST (red)

(T °C); (B) relative humidity (RH %) with dashed line as mean RH; (C) wind speed ($u\ ms^{-1}$);

and (D) wind direction (WD °); at South-Pullu (4727 m a.s.l.) in the upper Ganglass catchment,

Leh from 1 September 2015 to 31 August 2017.

The daily measured total precipitation at the study site equals 97.8 and 153.4 mm w.e. during

the years 2015–16 and 2016–17 respectively. After adding 23% under catch (Thayyen et al.,

2015) [unpublished work] to the total snow measurements, the total precipitation amount equal

to 120.3 and 190.6 mm w.e. for the years 2015–16 and 2016–17 respectively. During the study

period, the observed highest single-day precipitation was 20 mm w.e. recorded on 23

September 2015 and the total number of precipitation days were limited to 63. The snowfall



occurs mostly during the winter period (December to March) with some years witnessing

extended intermittent snowfall till mid-June, as experienced in this study during the year 2016-

17.

The precipitation estimated by the ESOLIP approach at the study site equals 92.2 and 292.5

mm w.e. during the years 2015–16 and 2016–17 respectively. The comparison between

observed precipitation (mm w.e.) and the one estimated by the ESOLIP approach is given in

(Table *S1*).

### 3.2    Observed radiation components and snow depth

The observed daily mean variability of different components of radiation, albedo and snow

depth from 1 September 2015 to 31 August 2017 at South-Pullu (4727 m a.s.l.) is shown in

Figure 3. Daily mean $SW_{in}$ varies between 24 and 378 W m$^{-2}$ (Table 2). Highest hourly

instantaneous short wave radiation recorded during the study period was 1358 W m$^{-2}$. Such

high values of $SW_{in}$ are typical of a high elevation arid-catchment (e.g., MacDonell et al.,

2013). Persistent snow cover during the peak winter period for both the years extending from

January to March resulted in a strong reflection of $SW_{in}$ radiation (Figure 3A). During most of

the non-free period, mean daily $SW_{out}$ radiation (Figure 3A) remain more or less stable below

100 W m$^{-2}$. Daily mean $SW_{out}$ varies between 2.4 and 262.6 W m$^{-2}$ with a mean value of 83.3

W m$^{-2}$ (Table 2). The daily mean $LW_{in}$ shows high variations and ranges between 109 and 345

W m$^{-2}$ with an average of 220 W m$^{-2}$ (Figure 3B, Table 2). Whereas $LW_{out}$ was relatively stable

and varied between 211 and 400 W m$^{-2}$ with an average of 308 W m$^{-2}$ (Figure 3B, Table 2).

The $LW_{out}$ shows higher daily fluctuations during the summer months as compared to the core

winter months. The daily mean $SW_{n}$ during the study period ranges between 2.5 and 319 W m$^{-2}$ with a mean value of 127 W m$^{-2}$. The $SW_{n}$ follows the pattern of $SW_{in}$ with higher values

during summertime and low values and relatively stable during winter (Figure 3C). The daily

mean $LW_{n}$ varies between -163 and 17 W m$^{-2}$. The $LW_{n}$ does not show any seasonality and





remain more or less constant with a mean value of -88 W m$^{-2}$ (Figure 3C). The mean daily

observed $R_n$ ranges from -80.5 to 227.1 W m$^{-2}$ with a mean of 39.4 W m$^{-2}$ (Table 2). During

both the years 2015–16 and 2016–17, the $R_n$ was high in summer and autumn but low in winter

and spring. When the surface was covered with the thick snow during January to early April in

2015–16 and during January to early May in 2016–17, the $R_n$ rapidly declined to low, or even

became negative values (Figure 3D). Albedo (α) is calculated as the ratio of SW$_{out}$ to SW$_{in}$ and

is of particular importance in the SEB and in the Earth's radiation balance that dictates the rate

of heating of the land surface under different environmental conditions (Strugnell and Lucht,

2001). The daily mean observed α at the study site ranges from 0.1 to 0.48, with a mean value

of 0.2 (Table 2). The daily mean α was low in summer and high in winter and increased

significantly when the ground surface was covered with snow (Figure 3E).

Both the years (2015–16 and 2016–17) experienced contrasting snow cover characteristics

during the study period (Figure 3F). The year 2015-16 experienced low snow as compared to

2016-17. During the 2015-16 year, the snowpack had a maximum depth of 258 mm on 30

January 2016, whereas during the 2016-17 year, the maximum was 991 mm on 07 April 2017.

The snow cover duration was 120 days during low snow year (2015-16) and 142 days during

the high snow year (2016–17). The site became snow-free on 27 April in 2016 and on 23 May

in 2017. Higher elevations of the catchment become snow-free around 15 July in 2016 while

the snow cover at glacier elevations persisted till 22 August in 2017. For both the year's snow

cover at lower elevations initiated by the end of December and the catchment experienced sub-

zero mean monthly temperatures since October.




Table 2 Summary of observed daily mean radiation components ($SW_{in}$, $SW_{out}$, $LW_{in}$ and $LW_{out}$), net radiation ($R_n$), surface albedo ($\alpha$), air temperature (Ta), precipitation (P), relative

humidity (RH), wind speed ($u$), wind direction (WD) and snow depth (h) for the study period

(1 September 2015 to 31 August 2017) at South-Pullu (4727 m a.s.l.).

| Variable | Units | Min. | Max. | Mean |
|---|---|---|---|---|
| $SW_{in}$ | W m$^{-2}$ | 24.1 | 377.8 | 210.4 |
| $SW_{out}$ | W m$^{-2}$ | 2.4 | 262.6 | -83.4 |
| $\alpha$ | - | 0.1 | 0.5 | 0.2 |
| $LW_{in}$ | W m$^{-2}$ | 109.0 | 344.7 | 220.4 |
| $LW_{out}$ | W m$^{-2}$ | 211.3 | 400.0 | -308.0 |
| $SW_n$ | W m$^{-2}$ | 2.5 | 318.7 | 127.0 |
| $LW_n$ | W m$^{-2}$ | -163 | 17.1 | -87.6 |
| $R_n$ | W m$^{-2}$ | -80.5 | 227.1 | 39.4 |
| $T_a$ | °C | -19.5 | 13.1 | -2.5 |
| $u$ | m s$^{-1}$ | 0.6 | 7.1 | 3.1 |
| WD | [°] | 28 | 245 | 148 |
| RH | % | 8 | 98 | 43.3 |
| P | mm w.e | 0 | 24.6 | 3.0 |
| h | mm | 0 | 991 | 102 |



Figure 3  Observed daily mean values of (A) incoming ($SW_{in}$) and outgoing ($SW_{out}$) shortwave

radiation, (B) incoming ($LW_{in}$) and outgoing longwave ($LW_{out}$) radiation, (C) net shortwave

($SW_n$) and longwave radiation ($LW_n$), and (D) net radiation ($R_n$), (E) surface albedo and (F)

snow depth (h, mm) at South-Pullu (4727 m a.s.l.) from 1 September 2015 to 31 August 2017.



### 3.3 Modelled surface energy balance

The mean daily variability of modelled surface energy balance (SEB) components is shown in

Figure 4. The sign convention adopted is as energy is considered as a gain for the surface or system if $R_n$ is positive and negative for H and LE. Conversely, energy is considered as loss for the surface or system, if $R_n$ is negative and positive for H and LE (Endrizzi, 2007; Oke, 2002). The average daily simulated $R_n$ ranges between -35.4 to 136.9 W m$^{-2}$ with a mean value of 28.9 W m$^{-2}$. The $R_n$ shows the seasonal variability and decreases as the ground surface gets

covered by seasonal snow cover during wintertime, and increases as the ground surface become snow-free (Figure 4A). From December to March of both the years (2015-16 and 2016-17), $R_n$ decreases and is negative during snow accumulation and remains close to zero during the melting time. For the rest of the time, $R_n$ remains positive. The daily mean H ranges between - 34.6 to 70.4 W m$^{-2}$ with a mean value of 13.5 W m$^{-2}$. The H is negative from mid-January to

April (2015-16) and January to June (2016-17) due to the presence of seasonal snow cover (Figure 4B). Rest of the period H remain positive and larger (~30 W m$^{-2}$) for most of the time. The seasonal variation in H points to a larger temperature gradient in summer than in winter. The daily mean LE ranges between -7.2 to 71 W m$^{-2}$ with a mean value of 12.8 W m$^{-2}$. During the snow-free freezing period (October to December) of both the years, the LE decreases (from

positive to zero) due to the freezing of moisture content in the soil and also fluctuates close to zero. Furthermore, when the seasonal snow is on the ground, the LE is positive, indicating sublimation and keeps increasing (more positive) after snowmelt indicating evaporation is taking place.

The heat conduction into the ground G remains relatively a smaller component in the SEB

(Figure 4C). The mean daily G ranges between -34.5 to 67.2 W m$^{-2}$ with a mean value of 0.4 W m$^{-2}$. The sign of the G, which shifted from positive during summer to negative during winter, is a function of the annual energy cycle. The heat flux available at the surface for melting ($F_{surf}$)





ranges between -21.8 to 77.1 W m$^{-2}$ with a mean value of 2.1 W m$^{-2}$ (Table 3). During the

summer, when snow melting conditions were prevailing, the $F_{surf}$ turns positive as a result of

energy available for melt (Figure 4C). Otherwise, the $F_{surf}$ become zero.

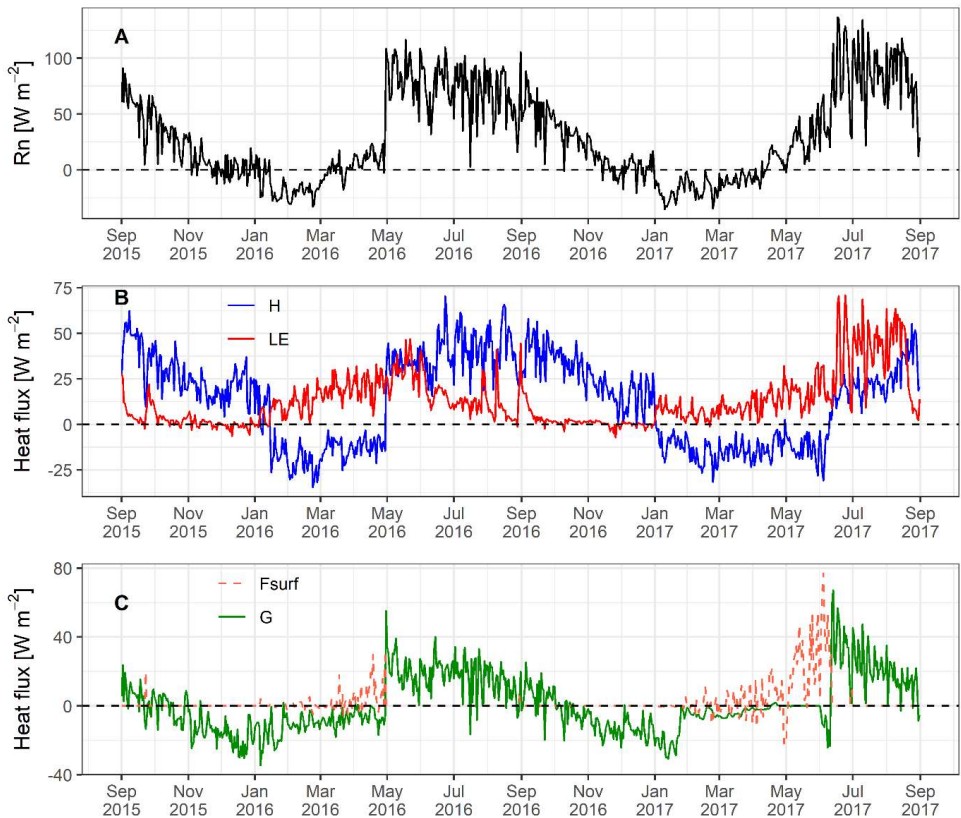

Figure 4 GEOtop simulated daily mean values of surface energy balance components (A) net

radiation (Rn), (B) sensible (H) and latent (LE) heat flux, (C) ground heat flux (G) and surface

heat flux ($F_{surf}$) at South-Pullu (4727 m a.s.l.) from 1 September 2015 to 31 August 2017.




Table 3 Mean daily range of GEOtop simulated SEB (W m$^{-2}$) components for the study period (1 September 2015 to 31 August 2017) at South-Pullu (4727 m a.s.l.).

| Variable | Min. | Max. | Mean |
|---|---|---|---|
| $R_n$ | -35.4 | 136.9 | 28.9 |
| H | -34.6 | 70.4 | 13.5 |
| LE | -7.2 | 71.0 | 12.8 |
| G | -34.5 | 67.2 | 0.4 |
| $F_{surf}$ | -21.8 | 77.1 | 2.1 |


The diurnal variation of modelled SEB components ($R_n$, LE, H and G) for the 2015–16 and 2016–17 years are shown in Supplementary Figures *S3* and *S4*. Clear sky days like 14 December 2015 (pre-winter), 18 February 2016 (winter), 30 April 2016 (post-winter), and 31 August 2016 (summer) were selected for the 2015–16 year (Figure *S3*). Similarly, for the 2016–

17 year, the following representative days were selected: 14 December 2016, 23 February 2017, 28 April 2017, and 17 August 2017 (Figure *S4*). During the representative days of both the years, the significant difference was the changing amplitude of the energy fluxes.

In the 2015–16 year (Figure *S3*), the amplitude of $R_n$ and the G during pre-winter, post-winter and summer season were the largest and smallest in winter. The G peaks earlier than those of

the LE and H during the pre-winter, post-winter and summer season. The LE and H show strong seasonal characteristics such as (a) during pre-winter season, the magnitude of diurnal variation of H was greater than LE depicting lesser soil moisture content because of freezing conditions at that time, (b) during winter season, the amplitude of LE was slightly greater (sublimation process) than H, (c) during post-winter of low snow year, the amplitude of LE and H was more

or less equal and (d) during summer season, the amplitude of LE was greater (higher soil moisture content and evaporation rate) and comparable to H, which is opposite to that of the pattern seen during pre-winter season. However, during winter season the G was close to zero.



The $R_n$ and G increased rapidly after the sunrise and changed the direction from negative to positive during pre-winter, post-winter and summer seasons. After sunset, the $R_n$ and G again

became negative rapidly, but the LE and H gradually decreased to lower values. The LE and H in the morning increased 1 to 2 hours after the $R_n$ during pre-, post-winter and summer season. In the 2016–17 year (Figure *S4*), the pre-winter, winter and summer were the same as that of the 2015–16 year. However, during the post-winter season of the 2016–17 year (28 April 2017) the main difference in diurnal changes were found because of the extended snow cover till May

during that year. The amplitude of Rn was slightly larger. However, all other components (LE, H and G) were of almost zero amplitude.

### 3.4 Comparison of seasonal distinction of SEB during low and high snow years

The seasonal distinction of modelled SEB components ($SW_n$, $LW_n$, $R_n$, LE, H, G and $F_{surf}$) for the low and high snow years of the study period is analysed. The seasons were defined as winter

(Sep-April) and summer (May-Aug) (Table 4). These seasons were further divided into two sub-seasons each such as winter but without snow (Sep, Oct, Nov and Dec) and winter with snow (Jan, Feb, Mar and Apr). Similarly, the summer season was divided into two sub-seasons called early summer (May and June; some years with extended snow) and peak summer (July and August).

Table 4 Mean seasonal values of modelled surface energy balance components.

| SEB Components [$W\,m^{-2}$] | 2015-16 | | | | 2016-17 | | | |
|---|---|---|---|---|---|---|---|---|
| | Winter (Sep to Apr) | | Summer (May to Aug) | | Winter (Sep to Apr) | | Summer (May to Aug) | |
| | Sep to Dec (Non-Snow) | Jan to Apr (Snow) | May to Jun (Non-Snow) | Jul-Aug (Peak Summer) | Sep to Dec (Non-Snow) | Jan to Apr (Snow) | May to Jun (Extended Snow) | Jul-Aug (Peak Summer) |
| $SW_n$ | 131.5 | 65.3 | 184.1 | 170.7 | 134.4 | 50.8 | 109.6 | 176.5 |
| $LW_n$ | -107.9 | -70.2 | -104.0 | -102.6 | -111.3 | -61.2 | -61.7 | -91.6 |
| $R_n$ | 23.6 | -4.9 | 80.1 | 68.1 | 23.1 | -10.4 | 47.9 | 84.9 |
| LE | 2.6 | 15.0 | 23.6 | 11.0 | 1.8 | 8.9 | 23.4 | 39.0 |
| H | 27.9 | -12.4 | 36.1 | 42.1 | 27.1 | -13.6 | -3.4 | 26.8 |
| G | -7.1 | -9.4 | 20.4 | 14.9 | -5.7 | -6.9 | 9.3 | 19.1 |
| $F_{surf}$ | 0.2 | 1.9 | 0.0 | 0.1 | 0.0 | 1.2 | 18.6 | 0.0 |



The mean seasonal variability of modelled SEB during the summer and winter seasons is reported in Table 4. The SEB results show that the mean seasonal $SW_n$ was highly variable as expected. Both the years observed comparable $SW_n$ during the snow-free winter period.

However, the $SW_n$ was comparatively smaller (50.8 W m$^{-2}$) during the snow sub-season of 2016-17 as compared to 2015-16 (65.3 W m$^{-2}$). Similarly, comparable $SW_n$ was observed during the peak summer sub-season of both the years. The key difference is seen during the early summer season (May-June) of 2016 and 2017. 2015-16 was a low snow year with early snowmelt out, and 2016-17 was a high snow year with extended snow cover and late snowmelt

out. The $SW_n$ of early summer period was correspondingly smaller (109.6 W m$^{-2}$) during 2017 as compared to 2016. The $LW_n$ shows less seasonal variability. Both the years observed comparable $LW_n$ during the winter season. However, the $LW_n$ was comparatively smaller during the extended snow (-61.7 W m$^{-2}$) and peak summer (-91.6 W m$^{-2}$) sub-season of 2016-17 as compared to the May-June (-104.0 W m$^{-2}$) and peak summer (-102.6 W m$^{-2}$) sub-seasons

of 2015-16. The $R_n$ (given by the sum of $SW_n$ and $LW_n$) followed the pattern of $SW_n$. Both the years observed comparable $R_n$ during the snow-free winter period. However, the $R_n$ was comparatively smaller (-10.4 W m$^{-2}$) during the snow sub-season of 2016-17 as compared to 2015-16 (-4.9 W m$^{-2}$). The key difference of $R_n$ is observed during early summer (May-June) and peak summer (Jul-Aug) of 2016 and 2017, respectively. The $R_n$ of the early summer period

was correspondingly larger (80.1 W m$^{-2}$) during 2016 as compared to 2017 (47.9). However, an opposite pattern was observed during the peak summer sub-season.

The LE flux shows seasonal variability for both the years. Both the years observed comparable LE during the winter season and May-June sub-season. The key difference in LE is observed during peak summer sub-season (Jul-Aug) of 2016 and 2017. In the peak summer sub-season

of 2016-17, the LE was higher (39 W m$^{-2}$) as compared to the 2015-16 (11 W m$^{-2}$). The reason behind this is due to the lesser amount of soil water content availability for evaporation during



2015-16 in comparison to high snow year 2016-17. The larger LE during the snow sub-season of both the years shows that sublimation is taking place. The H flux also shows the seasonal variability. The H was comparable during the winter season of both the years. During the

summer peak summer sub-season of the 2015-16 year, the H was slightly larger (4.1 W m$^{-2}$) as compared to 2016-17 (26.8 W m$^{-2}$). The key difference in H was observed during the extended snow sub-season of the 2016-17 year when H was much smaller (-3.4 W m$^{-2}$) compared to 2015-16 (36.1 W m$^{-2}$). The reason is due to the extended snow cover during the 2016-17 year.

During the winter season of both the years, the G was negative and changed the sign to positive during the summer season. The G was comparatively a smaller component. The mean seasonal $F_{surf}$ was almost equal to zero during all the seasons except during the snow sub-season of both the years and extended snow sub-season of the 2016-17 year. The $F_{surf}$ (heat flux available for melt) was much higher (18.6 W m$^{-2}$) during the extended snow sub-season of the 2016-17 year.

From the inter-year comparison, it was found that the extended snow sub-season of the 2016-17 (high snow year) showed the major differences in energy fluxes between the years.

### 3.5 Model evaluation

In this section, the capability of the 1D GEOtop model to reproduce the point-scale SEB is evaluated. The model was evaluated based on observed radiation components, snow depth and

one-year GST. In this study, the simulation results are based on the standard model parameters obtained from the literature (Table 2 and 3, Gubler et al., 2013) and were not improved by trial and error and the same simulation results are used for model evaluation.

### 3.5.1 Evaluation of radiation components

The first step in our model evaluation was to test the radiation components estimated by the

model. The comparison of two-year hourly simulated radiations components $SW_{in}$, $SW_{out}$, $LW_{in}$ and $LW_{out}$ against the field observation are shown in Figure 5. The observed and GEOtop





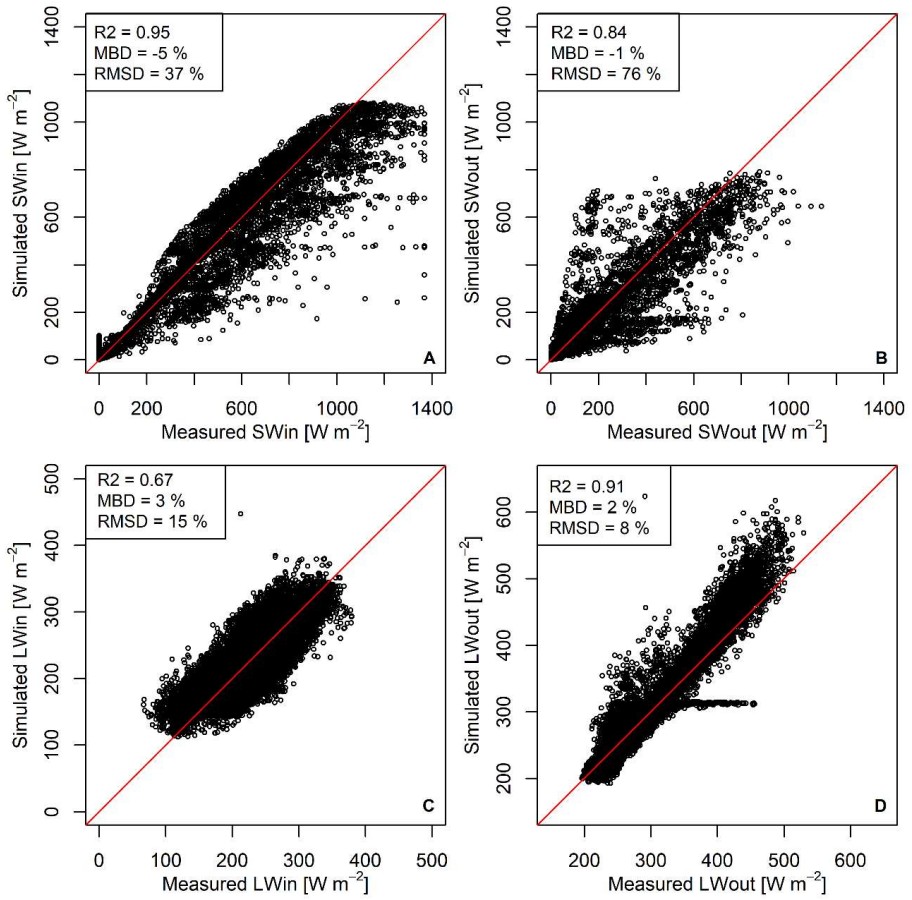

Figure 5 Scatter plots of hourly observed and simulated (A) incoming shortwave radiation (SW$_{in}$), (B) outgoing shortwave radiation (SW$_{out}$), (C) incoming longwave radiation (LW$_{in}$)

and (D) outgoing longwave radiation (LW$_{out}$) at South-Pullu (4727 m a.s.l.) from 1 September 2015 to 31 August 2017. The solid red lines are the 1:1 lines.

estimated SW$_{in}$ showed a strong linear relationship (R$^2$ = 0.95) and was slightly underestimated (MBD = -5 %) with a high RMSD value of 37 % (Figure 5A). The GEOtop simulated SW$_{in}$ fulfils the criteria of −5% ≤ MBD ≤ 5% set by Badescu et al. (2012) for estimation of global

SW$_{in}$ for the Iqbal (1983) model, but the criteria of RMSD ≤ 15% is not fulfilled. The SW$_{out}$



also shows good linear relationship ($R^2$ = 0.84) but it is slightly underestimated (MBD = -1 %) with high RMSD value of 76 % (Figure 5B). The $LW_{in}$ does not show a good linear relationship ($R^2$ = 0.67) and was slightly overestimated (MBD = 3 %) with RMSD value of 15 % (Figure 5C). The $LW_{out}$ shows a good linear relationship ($R^2$ = 0.91) but the GEOtop slightly

overestimates the $LW_{out}$ (MBD = 2 %) with RMSD value of 8 % (Figure 5D).

### 3.5.2 Evaluation of snowpack

The main objective of the evaluation of snowpack build up and melt out is to assess the robustness of the modelling exercise. Snow depth variations simulated by GEOtop are compared with observations from 1 September 2015 to 31 August 2017 (Figure 6). The model

captures the peaks, start and melt-out dates of the snowpack, as well as overall fluctuations ($R^2$ = 0.93). The maximum h simulated by the GEOtop was 1023.7 mm in comparison to the 1020 mm measured in the field. In the low snow year, the maximum simulated h was 290 mm in comparison to the 280 mm measured in the field. During the melting period of the low and high snow years, the snow depth was slightly under-estimated. However, during the

accumulation period of high snow year (2016-17), the h was rather overestimated.

Furthermore, the simulated melt-out date during the 2016-17 year was slightly delayed by a few days. Overall during the study period, the h was slightly overestimated with MB value of 6 mm. The RMSE value between observed and simulated h was 55 mm. The scatter plot showing the comparison between the simulated and measured h is given in supplementary

material (Figure *S5*).

Furthermore, the performance of the ESOLIP estimated precipitation was evaluated by running the GEOtop model twice, (a) first model run was made with precipitation data measured in the field, and (b) second model run was made with the ESOLIP estimated precipitation as input. The difference in the simulation of snow melt-out date shows the overall quality of the

precipitation data. In comparison to the observed precipitation data as model input, the timing,



evolution of snowpack and its melt-out are very well reproduced by the GEOtop model when

ESOLIP estimated precipitation was used as input (Figure 6). Hence, we can say that the

ESOLIP is a good approach for precipitation estimation, where snow depth and basic

meteorological measurements are available.

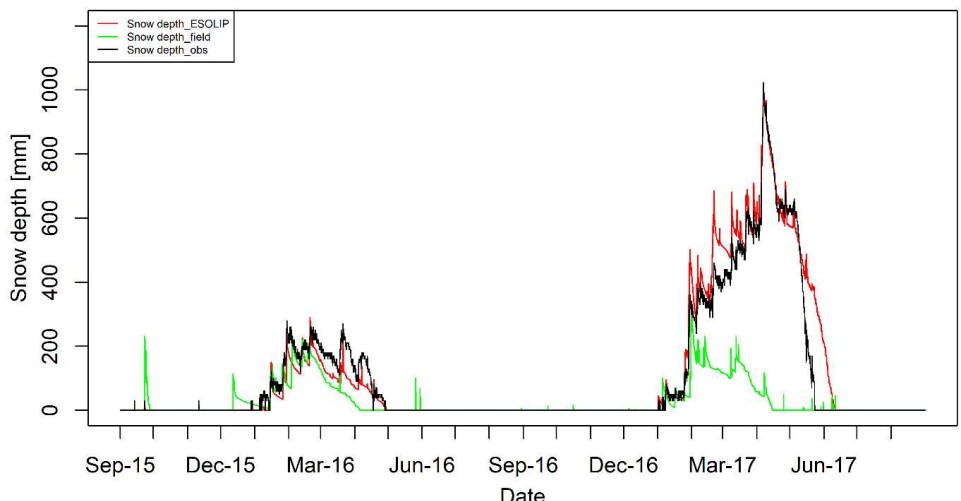

Figure 6 Comparison of hourly observed and GEOtop simulated snow depth (h, mm) at South-

Pullu (4727 m a.s.l.) from 1 September 2015 to 31 August 2017. The black line denotes the

snow depth measured in the field by SR50 sensor. The red (Snow depth_ESOLIP) and green

(Snow depth_field) lines in the plot denote the GEOtop simulated snow depth based on

ESOLIP estimated precipitation and precipitation measured in the field respectively.

### 3.5.3 Evaluation of near-surface ground temperatures (GST)

After evaluating the SEB and the snow cover, we evaluated the simulated GST at the study site

using GEOtop. GST is simulated (GST_sim) on an hourly basis and compared with the

observed values (GST_obs) near the AWS, available from 1 September 2016 to 31 August

2017 (Figure 7). The results show a fairly good linear agreement between the simulated and

observed GSTs (Figure *S6*, $R^2$ = 0.95). The GST was slightly underestimated with the MB

value of -0.76 °C. The high RMSE value (1.8 °C) shows large outliers in the estimation of



GST. However, the thermal influence of the winter snowpack was simulated reasonably well by the 1D GEOtop. For example, the start of dampening of temperature fluctuations by the

snowpack was reasonably well estimated by the model; however, towards the melt-out of the snowpack, the simulated zero-curtain was little longer as compared to the observation. The possible reason for this behaviour might be due to the estimated precipitation used as input to the model.

In conclusion, given that the GEOtop can simulate the point snow depth and the GST properly,

hence, we believe that the model is reliable enough to make robust calculations of SEB in complex topographies.

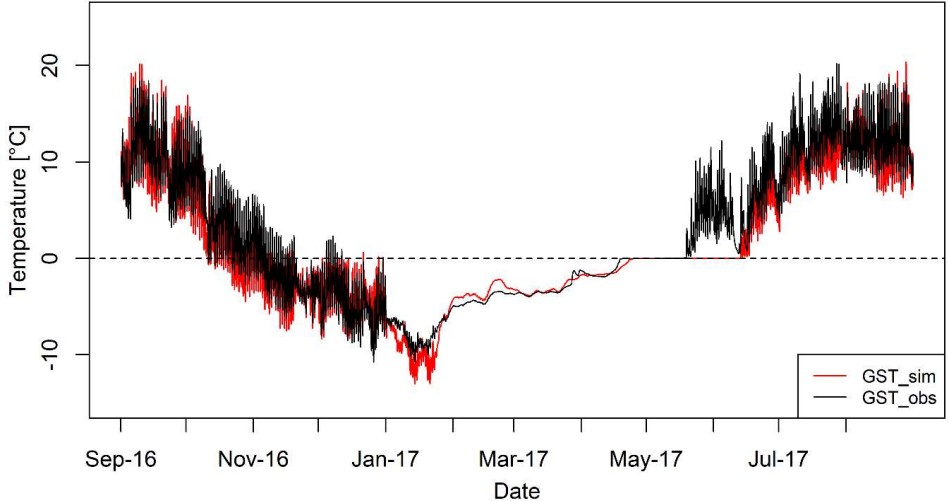

Figure 7 Comparison of hourly observed (GST_obs) and GEOtop simulated near-surface ground temperature (GST_sim, °C) at South-Pullu (4727 m a.s.l.) from 1 September 2016 to

31 August 2017.



## 4 Discussion

### 4.1 A distinction of SEB variations during low and high snow years

Realistic reproduction of seasonal and inter-annual variations in snow depth during the low (2015–16) and high snow (2016–17) years points towards the credible simulation of the SEB during the study period. We further investigated the response of SEB components during these years with contrasting snow cover for a better understanding of the critical periods of meteorological forcing and its characteristics.

To understand the critical periods of meteorological forcing and its effect on modelled SEB fluxes, we will discuss the diurnal variation of modelled SEB only for one season, i.e., post-winter, which shows major differences in the amplitude of energy fluxes (Figure 8). During the pre-winter, winter and summer seasons (Figure *S3*, *S4*), the diurnal variations of the SEB fluxes on representative, clear days for the 2015-16 year were similar compared to 2016-17. However, during the post-winter season of both the years (Figure 8), the SEB fluxes show different diurnal characteristics. During the post-winter season of 2016–17 year (28 April 2017) the main difference in diurnal changes were found because of the extended snow cover till May during that year. For the 2016–17 year, the amplitude of $R_n$ was slightly larger, whereas, all other components (LE, H and G) were of almost zero amplitude (Figure 8B). The smaller amplitude of LE, H and G is due to the smaller input (solar radiation) and the extended seasonal snow on the ground. Therefore, we can say that the different SEB characteristics during these two years' is a reaction to the forcing of precipitation via snowfall.

During the study period, the proportional contribution shows that the net radiation component dominates (80%) the SEB followed by H (9%) and LE fluxes (5%). The G was limited to 5% of the total flux, and 1% was consumed for melting the seasonal snow. The percentages of the energy fluxes were calculated by following the approach of Zhang et al. (2013).





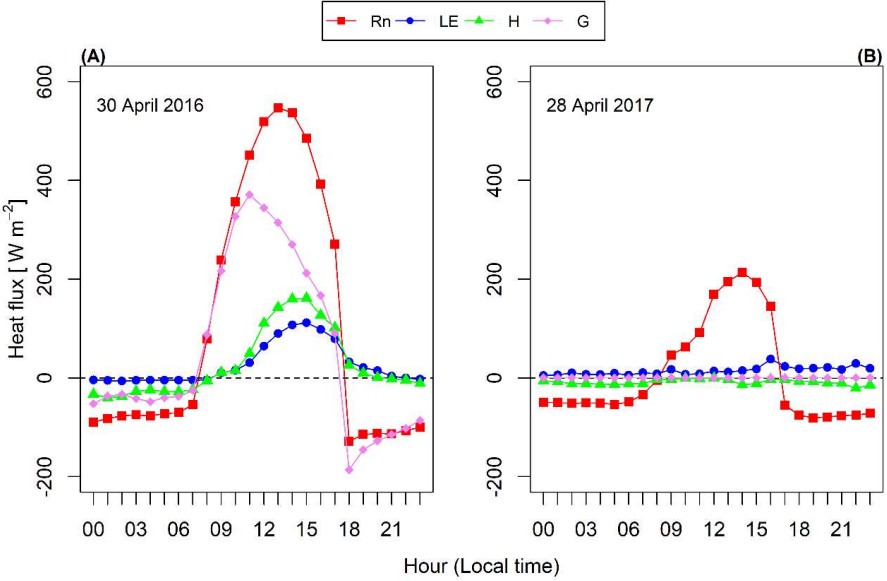


Figure 8 The diurnal change of GEOtop modelled surface energy fluxes on (A) 30 April 2016, and (B) 28 April 2017 representing post-winter season for the 2015-16 and 2016–17 years respectively at South-Pullu (4727 m a.s.l.), in the upper Ganglass catchment, Leh.

The comparison of percentages of modelled SEB components during low (2015-16) and high

(2016-17) snow years show that net radiation component was equal to 81 and 80% for 2015-16 and 2016-17, respectively. However, the LE was lower (5%) during a low snow year in comparison to 6% modelled during the high snow year. The H was 9 and 8% during the low and high snow years, respectively. The G was found to be 4.7 and 4.5% for the 2015-16 and 2016-17 years, respectively. During the low and high snow years, the energy utilised for

melting seasonal snow was 0.4 and 1.7%, respectively. The mean monthly modelled SEB components for both the years are given in Table *S2*.

Furthermore, during the study period, the partitioning of energy balance show that 47% (13.5 W m$^{-2}$) of R$_n$ (28.9 W m$^{-2}$) was converted into H, 44% (12.8 W m$^{-2}$) into LE, 1% (0.4 W m$^{-2}$) into G and 7% (2.1 W m$^{-2}$) for melting of seasonal snow. During the 2015-16 year, the 58%



(18.2 W m$^{-2}$) of $R_n$ (31.3 W m$^{-2}$) was consumed by H, 37% (11.6 W m$^{-2}$) by LE, 1% (0.4 W m$^{-2}$) by G and 3% (1.1 W m$^{-2}$) was used for melting seasonal snow, whereas during the 2016-17 year, 31% (8.4 W m$^{-2}$) of $R_n$ (26.9 W m$^{-2}$) was consumed by H, 52% (14.0 W m$^{-2}$) by LE, 2% (0.5 W m$^{-2}$) by G and 15% (4.0 W m$^{-2}$) was used for melting seasonal snow. However, a distinct variation of energy flux is observed during the month of May-June, when one of the years

(2016-17) experienced extended snow. The mean monthly comparison of modelled SEB fluxes during the 2015–16 and 2016–17 is shown in (Figure 9). In the early winter (September to December), the mean monthly $R_n$ and LE were, respectively, the same for both the years. The main difference can be seen from March to May (Figure 9A) for Rn, and LE from February to May (Figure 9B) when the $R_n$ and LE during low snow year were larger than the high snow

year. During both the years, the mean monthly H was more or less equal from September to April (Figure 9C), but during the low snow year, the H was higher for rest of the months. However, the mean monthly G does not show a clear pattern (Figure 9D) but the modelled G during both the years follows the pattern of $R_n$. We found that April, May and June are the most critical months by experiencing higher $R_n$, LE, and H during low snow year as compared

to high snow years (Figure 9). Also, we found that the mean annual $R_n$, LE, H and G were larger during low snow years as compared to high snow years.

In both low and high snow years (Figure 9E, F), the mean monthly H and LE heat fluxes show a seasonal characteristic, such as the H was higher than the LE from September to December for both the years. But for the low snow year, the H was lower than the LE from January to

April and after that from May to August again the H was higher than LE. In the high snow year, the LE was higher than the H from January to July and was positive pointing towards the sublimation (January to May) and evaporation processes (June to August).



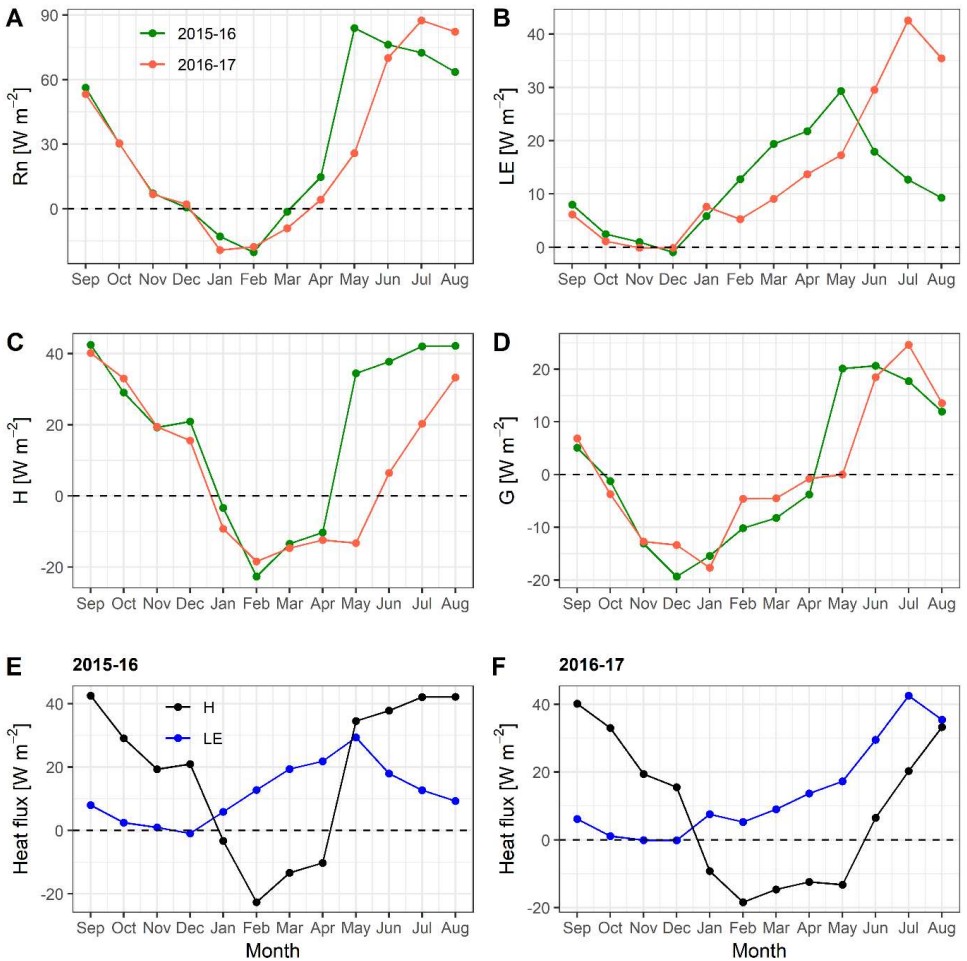

Figure 9 Comparison of estimated mean monthly surface energy balance components (W m$^{-2}$) (A) R$_n$, (B) LE, (C) H, and (D) G for the low (2015-16) and high (2016-17) snow years, and (E) and (F) represent the mean monthly surface H and LE (W m-2) for low (2015-16) and high (2016-17) snow years, respectively, at South-Pullu (4727 m a.s.l.).

In early October, the LE began to weaken up to the December for both the years as the seasonally frozen ground began to freeze. Therefore, the seasonal freezing/thawing of the ground affect the LE causing its rapid decrease/increase. Early in October/November, when the seasonally frozen ground began to freeze, the H did not show any significant variability.



However, during summer (from April onwards in 2015–16 and from May onwards in 2016–17), after the snowmelt, the H increases significantly. Similar kind of variability in the LE and

H is also reported from the seasonally frozen ground and permafrost regions of the Tibetan plateau (Gu et al., 2015; Yao et al., 2011).

### 4.2    Influence of snow cover on near-surface ground temperature

Snow cover affects the ground thermal regime by altering the surface energy balance due to its unique characteristics such as, (a) high albedo, (b) high absorptivity, (c) low thermal

conductivity, and (d) high latent heat due to snowmelt that is a heat sink (Goodrich, 1982; Gruber, 2005; Zhang, 2005). In comparison to other natural ground surface materials, the low thermal conductivity allows the snow cover to act as an insulator between the atmosphere and the ground. To analyse the effects of snow cover on GST, we plotted the relationship between observed snow depth and GST during the seasonal snow period from 1 January to 23 April

2017 at South-Pullu (4727 m a.s.l.) (Figure 10). For the shallow snow depth, the GST was smaller, and as the depth of snowpack starts increasing, the GST also starts increasing towards 0 °C. The GST varied from -10 °C to about -2 °C under 40 and 900 mm of snow, respectively. During the low snow year, the modelled snow depth show no or small insulating effect on GST (Figure *S7*). However, during the high snow year, the variations of GST are dampened with

increasing snow depth. Furthermore, during the high snow year, only the modelled snow depth greater than 350 mm shows an insulating effect on GST.

The timing of snow cover start and its duration has a non-linear influence on the ground surface temperatures (Bartlett et al., 2004). In the early winter, a thin snow cover can cool the ground, whereas a thick snow cover insulates the ground from cold air temperature variations (Keller

and Gubler, 1993). During both the years, the snowfall in the catchment occurred by the last week of December facilitating the ground cooling by almost three months (October to December) of sub-zero temperatures up to -20 °C. This could be a key factor in controlling the

thermal regime of permafrost in the area. Extended snow cover during the high snow year

insulates the ground from warmer temperature during May. Therefore, the observed snow

cover processes in the study area during these two years have a positive influence on permafrost

at the study site. However, long-term studies and larger number of direct snow depth

measurements at different locations are required for a better understanding of these processes.

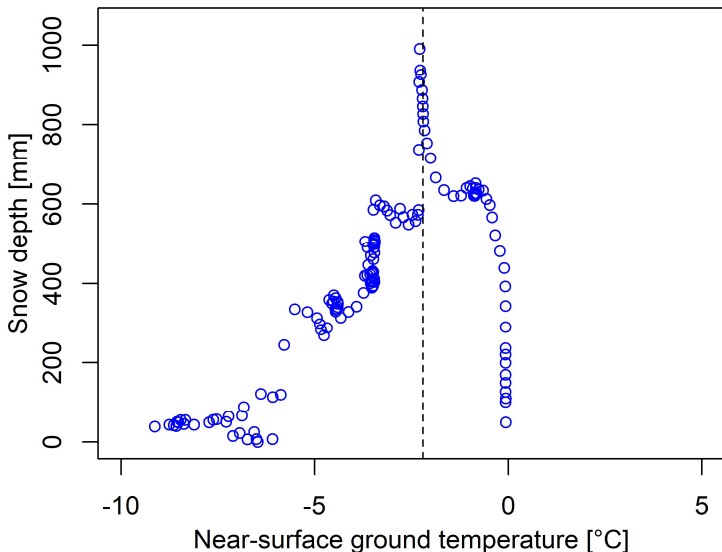

Figure 10 Relationship between daily mean observed snow depth (mm) and near-surface

ground temperature (° C) from 1 January to 23 April 2017 at South-Pullu (4727 m a.s.l.). The

vertical dashed line shows the peak of the snow accumulation, and after that, snowpack starts

melting.

### 4.3    Comparison with other environments

In this section, the SEB components from our cold-arid catchment in Ladakh, India are

compared with other cryospheric systems, globally (Table 5). Although aiming to represent

differing permafrost environments, this comparison also includes SEB studies on glaciers for

lack of other data. In most of the studies referred here, the radiation components are measured,

and the turbulent (H and LE) and ground (G) heat fluxes are modelled.





The mean $SW_{in}$ measured (210 W m$^{-2}$) in this study site was comparable with values reported

from the Tibetan Plateau (200 W m$^{-2}$) but lower than the Andes (239 W m$^{-2}$) and significantly

higher than the values reported from other studies such as the Alps (136 W m$^{-2}$). The different

surface albedo ($\alpha$) values help to distinguish the surface characteristics. The mean $\alpha$ for all the

sites where radiation balance is measured either on bedrock or tundra vegetation was smaller

than those measured over firn or ice during summer with few exceptions. Albedo ranges for

glacier ice from 0.49 to 0.69 and for tundra/bedrock from 0.25 to 0.54. The mean $LW_{in}$ (220.4

W m$^{-2}$) observed at Leh station is comparable with values observed at Tibetan Plateau (221 W

m$^{-2}$) and smaller than the other studies except for Antarctica (184.1 W m$^{-2}$). In the present

study, the mean $SW_n$ is the largest source of energy gain (127 W m$^{-2}$) and $LW_n$ the largest

energy loss and strongly negative (-87.6 W m$^{-2}$) and both were higher than in other studies.

However, the $SW_n$ (127 W m$^{-2}$) observed at Leh station was comparable with the values

observed in the tropical Andes (123 W m$^{-2}$). The mean measured RH (43 %), and the mean

annual precipitation are both smaller in this study than in the other areas compared.

Based on the comparison of measured radiation and meteorological variables with other, better-

investigated regions of the world (Table 5), it was observed that our study area have a very low

RH (40% compared to ~70% in the Alps) and cloudiness, leading to (a) Reduced $LW_{in}$ and

strongly negative $LW_n$ (~90 W m$^{-2}$ on average, much more than in the Alps). This will lead to

surfaces being overall colder than at a similar location with more RH, (b) Increased $SW_{in}$. This

will mean that sun-exposed slopes will receive more radiation and shaded ones less (less diffuse

radiation) than in comparable areas, and (c) Increased cooling by stronger evaporation in wet

places such as meadows. Where there is enough water, you can cool the ground significantly.





Table 5 Comparison of mean annual radiation and SEB components and meteorological variables with other regions of the world. ($SW_{in}$ = Incoming shortwave radiation, $SW_{out}$ = Outgoing shortwave radiation, albedo = $\alpha$, $LW_{in}$ = Incoming longwave radiation, $LW_{out}$ = Outgoing longwave radiation, $SW_n$ = Net shortwave radiation, $LW_n$ = Net longwave radiation, RH = Relative humidity, $R_n$ = Net radiation, LE = Latent heat flux, H = Sensible heat flux, G = Ground heat flux, SEB = energy available at surface, MAAT = Mean annual air temperature, P = Precipitation, NA = Not available). The LE, H, and G are the modelled values. All the radiation components and heat fluxes are in units of W m$^{-2}$.

| Variable | Leh | Tibetan Plateau | Swiss Alps | | Tropical Andes | New Zealand (Alps) | Canada | Sub-Arctic | Greenland | High Arctic (Norway) | | | | Antarctic | |
|---|---|---|---|---|---|---|---|---|---|---|---|---|---|---|---|
| $SW_{in}$ | 210.4 | 230 | 136 | 149 | 239 | 140 | 136 | 101.3 | 110 | 79.5 | 122 | 78 | 108 | 124 | 94.2 |
| $SW_{out}$ | -83.4 | -157 | -72 | -74 | -116 | -93 | -94 | -25.7 | -70 | -39.5 | -38 | -42 | -70 | -79.7 | -52.0 |
| $\alpha$ (-) | 0.40 | 0.68 | 0.53 | 0.5 | 0.49 | 0.66 | 0.69 | 0.25 | 0.64 | 0.50 | 0.31 | 0.54 | 0.65 | 0.64 | 0.55 |
| $LW_{in}$ | 220.4 | 221 | NA | 260 | 272 | 278 | 248 | 310 | 246 | 263.7 | 261 | 254 | 272 | NA | 184.1 |
| $LW_{out}$ | -308.0 | -277 | NA | -308 | -311 | -305 | -278 | -349.8 | -281 | -299.0 | -300 | -286 | -292 | NA | -233.2 |
| $SW_n$ | 127.0 | 73 | 64 | 75 | 123 | 48 | 42 | 75.6 | 40 | 40.0 | 84 | 36 | 38 | 44.3 | 42.2 |
| $LW_n$ | -87.6 | -56 | -36 | -48 | -39 | -27 | -30 | -39.8 | -36 | -35.3 | -39 | -32 | -20 | -49.2 | -49.1 |
| RH (%) | 43.3 | 59 | 64 | 59 | 81 | 78 | 71 | ~75 | 75 | 74.8 | 83 | 74 | 77.9 | 50.8 | 69.4 |
| $R_n$ | 39.4 | 17 | 28 | 27 | 84 | 21 | 12 | 37.1 | 4 | 4.78 | 45 | 4 | 18 | -4.9 | -6.9 |



| | | | | | | | | | | | | | | | |
|---|---|---|---|---|---|---|---|---|---|---|---|---|---|---|---|
| **LE** | 12.8 | -11 | 6 | -1 | -27 | 1 | -15 | NA | NA | NA | NA | 6.8 | 1 | -62.1 | -5.0 |
| **H** | 13.5 | 13 | 36 | -3 | 21 | 30 | -5 | 2.9 | NA | NA | -34.2 | -6.9 | 15 | 28 | 12.1 |
| **G** | 0.4 | 2 | 3 | -2 | NA | 2 | 0.5 | 1.9 | NA | NA | -3.5 | ~0.5 | 3 | -0.12 | 0.2 |
| **MAAT (°C)** | -2.5 | -6.3 | 2.1 | -1.1 | 0.3 | 1.2 | -4.2 | 6 | -5.45 | -2.86 | -3.4 | -5.4 | -1.9 | -10.2 | -18.8 |
| **P (mm)** | 114 | 1250 | NA | NA | 970 | NA | NA | 369 | NA | 581.2 | 800 | NA | NA | NA | NA |
| **Time period** | Sep 2015 to Aug 2017 | Aug 2010 to Jul 2012 | Jan to Dec 2000 | Feb 1997 to Jan 1998 | Mar 2002 to Mar 2003 | Oct 2010 to Sep 2012 | 2002–2013 | Jan to Dec 2013 | Aug 2003 to Aug 2007 | Jan 2015 to Dec 2015 | Jan to Dec 2000 | Mar 2008 to Mar 2009 | Sep 2001 to Sep 2006 | Mar 2007 to Jan 2013 | Apr 1988 to Mar 1989 |
| **Surface type** | Bedrock/debris | Glacier ice | Glacier ice | Bedrock/debris | Glacier ice | Glacier ice | Glacier ice | Peatland | Glacier ice | Tundra vegetation | Bedrock/debris | Tundra vegetation | Glacier ice | Ice sheet | Ice sheet |
| **Location** | Cold-arid, Ladakh | Zhadang Glacier, Tibetan Plateau | Morteratschgletsche glacier, Switzerland | Murtèl- Corvatsch rock glacier, Switzerland | Antizana glacier 15, Ecuador | Brewster Glacier, New Zealand | Haig Glacier, Canadian rocky mountains | Peatland complex Stordalen, Sweden | west Greenland ice sheet | Bayelva, Spitsbergen, Norway | Juvvasshoe, southern Norway | Svalbard, Norway | Storbreen glacier, Norway | Schirmacher Oasis, Antarctica | Dronning Maud Land, Antarctica |
| **Elevation (m)** | 4727 | 5665 | 2100 | 2700 | 4890 | 1760 | 2665 | 380 | 490 | 25 | 1894 | 25 | 1570 | 142 | 1150 |





| Latitude | Source |
|---|---|
| 34.255° N | This Study |
| 30.476° N | (Zhu et al., 2015) |
| 46.400° N | (Oerlemans and Klok, 2002) |
| 46.433° N | (Stocker-Mittaz, 2002) |
| 0.467° S | (Favier, 2004) |
| 44.084° S | (Cullen and Conway, 2015) |
| 50.717° N | (Marshall, 2014) |
| 68.349° N | (Stiegler et al., 2016) |
| 67.100° N | (van den Broeke et al., 2008) |
| 78.551° N | (Boike et al., 2018) |
| 61.676° N | (Isaksen et al., 2003) |
| 78.917° N | (Westermann et al., 2009) |
| 61.600° N | (Giesen et al., 2009) |
| 70.733° S | (Ganju and Gusain, 2017) |
| 74.481° S | (Bintanja et al., 1997) |





## 5   Conclusion

In the high-elevation, cold–arid regions of Ladakh significant areas of permafrost occurrence are highly likely (Wani et al., 2020) and large areas experience deep seasonal freeze-thaw. The present study is aimed at providing first insight on the surface energy balance characteristics of this permafrost environment. The one-dimensional mode of GEOtop model was used to estimate the surface energy balance at South-Pullu (4727 m a.s.l.) in the upper Ganglass catchment from 1 September 2015 to 31 August 2017 using in-situ meteorological data. The model performance was evaluated using measured radiation components, snow depth variations and one-year near-surface ground temperatures which shows good agreement. For the period under study, the surface energy balance characteristics of the cold-arid site in the Indian Himalayan region show that the net radiation was the major component with a daily mean value of 28.9 W m$^{-2}$, followed by sensible heat flux (13.5 W m$^{-2}$) and latent heat flux (12.8 W m$^{-2}$), and the daily mean ground heat flux was equal to 0.4 W m$^{-2}$. During the study period, the partitioning of surface energy balance show that 47% (13.5 W m$^{-2}$) of $R_n$ (28.9 W m$^{-2}$) was converted into H, 44% (12.8 W m$^{-2}$) into LE, 1% (0.4 W m$^{-2}$) into G and 7% (2.1 W m$^{-2}$) for melting of seasonal snow. Among the two observation years, one was low snow year (maximum snow depth of 258 mm and duration of 120 days) and the another was high (maximum snow depth of 991 mm and duration of 142 days). During these low and high snow years, the energy utilised for melting seasonal snow was 3% and 15%, respectively. Key differences in surface energy balance characteristics were observed during early (May-June) and peak (July-August) summer season of the high snow year. For example, the latent heat flux was higher (39 W m$^{-2}$) during the peak summer of high snow year compared to the low snow year (11 W m$^{-2}$). However, the sensible heat flux during the early summer season of the high snow year was much smaller (-3.4 W m$^{-2}$) compared to the low snow year (36.1 W m$^{-2}$). The diurnal variation of surface energy balance components shows that the extended snowfall



during the high snow year affects surface energy balance characteristics at the study site. The air temperature throughout the study period varies between –23.7 to 18.1 °C with a mean annual average temperature of -2.5 °C, whereas, the near-surface ground temperature ranges from -

9.8 to 19.1 °C with a mean annual value of 2.1 °C. During both the low and high years, the snowfall in the catchment occurred by the last week of December facilitating the ground cooling by almost three months (October to December) of sub-zero temperatures up to -20 °C. The extended snow cover during the high snow year also insulates the ground from warmer temperature until May. Therefore, the late occurrence of snow and extended snow cover could

be the key factors in controlling the thermal regime of permafrost in the area. A comparison of observed radiation and meteorological variables with other regions of the world show that the study site/region at Ladakh have a very low relative humidity (RH) in the range of 43% compared to, e.g. ~70% in the Alps. Therefore, rarefied and dry atmosphere of the cold-arid Himalaya could be impacting the energy regime in multiple ways. (a) This results in the

reduced amount of incoming longwave radiation and strongly negative net longwave radiation, in the range of -90 W m$^{-2}$ compared to -40 W m$^{-2}$ in the Alps and therefore, leading to colder land surfaces as compared to the other mountain environment with higher RH. (b) Higher global shortwave radiation leads to more radiation received by sun-exposed slopes than shaded ones in comparable areas and wet places such as meadows, etc. experience increased cooling

as a result of stronger evaporation. However, sun-exposed dry areas could be warmer, leading to significant spatial inhomogeneity in permafrost distribution. The current study gives a first-order overview of the surface energy balance from the cold-arid Himalaya in the context of permafrost, and we hope this will encourage similar studies at other locations in the region, which would greatly improve the understanding of the climate from the region.




**Acknowledgements**

John Mohd Wani acknowledges the Ministry of Human Resource Development (MHRD) Government of India (GOI) fellowship for carrying out his PhD work. Renoj J. Thayyen thanks the National Institute of Hydrology (NIH) Roorkee and SERB (Project No.

EMR/2015/000887) for funding the instrumentation in the Ganglass catchment. The first insight into the use of GEOtop permafrost spin up scheme by Joel Fiddes is highly acknowledged. We acknowledge the developers of GEOtop, for keeping the software open-source and free. The source code of the GEOtop model 2.0 (Endrizzi et al., 2014) used is freely available at https://github.com/geotopmodel/geotop/tree/se27xx.

**Conflicts of interest**

The author(s) declare(s) that there is no conflict of interest.

**Author contributions**

JMW participated in data collection in the field, carried out the data analysis and processing, run the GEOtop model and prepared the manuscript. RJT arranged field instruments, organised

fieldwork for instrumentation and data collection, contributed to the data analysis and manuscript preparation. CSPO assisted in data analysis and manuscript preparation. SG assisted in setting up GEOtop model, analysis of results and manuscript preparation.



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
