# Peer review of "The surface energy balance in a cold-arid permafrost environment, Ladakh"

_The Cryosphere, 2019_

## Referee Comment (RC1) · Anonymous Referee #1 · 26 Mar 2020

Comments to the paper: The surface energy balance in a cold-arid permafrost environment, Ladakh, Himalaya, India John Mohd Wani, Renoj J. Thayyen, Chandra Shekhar Prasad Ojha, and Stephan Gruber

General comments:

This paper contains a two-year measurement series of surface energy balance (SEB) measurements at a high-altitude site in Ladakh, India and an application of an already well-tested energy balance model (GEOtop). Measured values such as the individual radiation components and the snow cover development are validated against the GEOtop model outputs. This paper represents an excellent, well-written and careful high-quality work, which is an important contribution to a better understanding of surface energy balance and heat fluxes in a very remote and high-altitude area, where not

many of these in situ measurements exist. Therefore, I would recommend this paper for publication after having considered some minor suggestions given below:

- Page 11: When presenting the energy balance equation, the authors use in my opinion a slightly confusing convention related to the flux directions. I would suggest that they use a more common convention very often used in cryospheric research that all fluxes towards the surface are positive and negative away from the surface, because the authors used a different convention often in the paper values are not clearly presented. As an example, in Table 2: the mean value of Sout is given as a negative value and the Min and Max values are given as positive values. The same is the case for LWout.

- Page 11: The authors present as their first objective on page 5: (a) quantify the point Surface Energy Balance (SEB)! When calculating the energy balance from measurements, it is then not clear, why the authors do not use their data to calculate the melt by using their measurements of the snow cover? I understand that they use the model to calculate the melt and also the ground temperatures with their model and use the measured data of snow cover and ground temperature as validation data. However, I have the impression that through this approach the authors mix different steps in the methodology and increase the degree of freedom unnecessarily. First, the authors may simply use all the available data to determine the MEASURED SEB based on the well-known and common approaches and then in a second step they make their model exercise, which is already very well done.

- Page 22, Table 2: the given albedo values seem to be not reasonable. The authors present for example (taken from Figure 3) measured SWin values in spring (April) of around 300 W m-2 and SWout values of 250 Wm-2. A corresponding value of albedo (alpha) would be higher than 0.5. Therefore, the max value of alpha should be higher. Please clarify!

- Table S2 in supp. material: I would recommend that you send your data to the Global

Energy Balance Archive (see also http://www.geba.ethz.ch and https://www.earth-syst-sci-data.net/9/601/2017/)

Specific comments: 1. Page 4, line 83: please cite here a text book such as Oke 1987 or Sellers 1965, because these are well-known facts since starting EB studies.

2. Page 10, line 218: what means 'controlled through parameters' -> please be more specific and explain more in detail.

3. Page 10, line 219: please delete s: ...mountain regions...

4. Page 11:, line 234: replace But with However,

5. Page 11, line 240/41: equation (4): why should LWn be only a function of Ts? Please delete the dependencies to Ts in equation, because further down the authors explicitly explain that these variables are not only depending on Ts.

6. Page 12, line 257-260: this is strongly dependent on the effective soil conditions, if you have rock surfaces it is completely different from fine sedimentary material. -> please clarify! Please explain in more detail the BATS, which is used here!

7. Page 14, line 296-298: what happens if your surface is bedrock?

8. Page 16, line 360: I would also like to see an evaluation of the turbulent heat fluxes!

9. Page 17, line 387: delete s: root mean square error

10. Page 20, Figure 2: would be nice to plot snow height in figure 2 A!

11. Page 21, line 468: what do mean with non-free? clarify!

12. Page 22, line 476: please reformulate the following sentence to: ...with higher values during summertime and low, relatively stable values during winter...

13. Page 22, line 481: please reformulate the following sentence to: ... with a thick snow cover during...

14. Page 22, line 483: please delete word: values

15. Page 23, Table 2: please control and adapt table 2 according to my comments under General Remarks.

16. Page 27, Table 3 and page 30 line 615: Fsurf values: please explain the signs of these values? Please also explain the variability of Fsurf in relation to your result outputs of your model? What is the meaning of Fsurf when it is negativ and there is no snow? Please clarify!

17. Page 27, line 615: please correct: available

18. Page 35, Figure 8: here it is important that most of the energy in Rn is used for melting (particularly in the year 2017) and this should be shown in the figure!

---

## Referee Comment (RC2) · Anonymous Referee #2 · 1 Jul 2020

Review of manuscript: "Towards a conceptual model of water routing for a debris-covered glacier" by Wani and co-authors submitted to The Cryosphere

This paper presents simulations of energy fluxes at the location of one automatic Weather Station at 4727m a.s.l. in the upper Ganglass catchment in the Ladakh region of India. Energy fluxes are calculated over a period of two years, from 1 Sept 2015 to 31 Aug 2017, using the GeoTop model. The model performance is evaluated against observations of radiative fluxes, snow depth and near surface ground temperature measured at 10cm from the surface. Then the authors analyse the energy fluxes obtained with the model and describe their temporal variability. They finally compare the average values obtained for each flux with values from the literature computed at other locations in the world, from the Tibetan Plateau to the Andes, irrespective of the

sites' elevation, the type of surface (ground, glacier surface, etc) as well as the models used for those simulations. The paper is interesting in that it presents energy fluxes at a remote location in a distinct climatic region of High Mountain Asia (HMA), dominated by very dry conditions and where permafrost has been identified as a dominant feature. The paper however is short, in its current form, of the quality needed for publication, both in terms of structure and language and in terms of content. I have several comments that the authors should address before the paper can be published. I was struck in particular by the lack of clear findings and a discussion of those beyond a simple description of the fluxes the authors obtain. In its current form the paper reads more like a report than a scientific paper. There is also a compelling lack of uncertainty analysis, which I would strongly encourage the authors to carry out (see one of my major comments below). Equally, the measurements are presented without any error or uncertainty assessment. I also have several comments on some of the methods used. The text contains many repetitions, it is redundant at times and on the other sides lacks key information (e.g. the values of the critical surface and soil properties used to run the model). The English needs a thorough proof-reading. In synthesis, this is an interesting paper that could represent a valuable contribution to energy balance studies, but needs very major revisions before can be accepted for publications in TC. I hope the authors will find my suggestions useful.

MAIN COMMENTS 1) English and style The English contains grammar errors (too many to detail here, but as an example often the third person plural is used when it should have been singular) and weird sentences. The writing style is often redundant and contains many repetitions – I have indicated some below. I had started suggesting corrections to the English but then stopped as this would take many pages and a lot of time. The paper however needs a careful and extensive proof-reading of both English and writing style, and the authors should make an effort to turn the manuscript into a more readable, polished and compelling paper. The abstract seems long and could be shortened and made more to the point. There are many repetitions in the paper, e.g. lines 242-243; 344-346; 511-512; and in many other instances. I would strongly

encourage the authors to go through the manuscript and polish/improve it substantially. In its current form, it is not appropriate for publication.

2) Paper structure I feel the paper structure needs to be improved in several places. First, I would suggest that the authors separate the study site and data section from the methods section, for readability. As it is now, they need many sub-sections to accommodate all this section content and this section is very long. Second, a lot of text that belongs to the Methods is contained in the Results section, to a point that the paper is extremely repetitive. Examples are on lines XXX Third, I would encourage the authors to reconsider the way their Results section is structured: first the observations are presented, then the energy fluxes described, and then those are validated with the observations. Before any discussion of the fluxes, they should be validated – otherwise we do not know on which we can have confidence and on which we can have less. I also have some major comments on the figures in this section, which are repetitive and do not make a very good use of space (see comment on Figures below). Finally, most of the content in the Discussion, and most of those figures, should actually be in the Results section, as they present the actual energy fluxes that are the main focus of this paper.

3) Aim of the paper It is not clear what the paper aim is. The authors state: "we aim to provide a foundation for better understanding the micro-climatological drivers affecting permafrost distribution and temperature regimes in the area, to build hypotheses about similarities and major differences with other, better-investigated permafrost areas". First, it does not seem that this study can contribute to understanding the drivers of permafrost distribution, given that it focuses on one single location. If however the authors think their analysis can contribute to this, they should devote the discussion to examine how their results about energy fluxes at one location can be relevant for permafrost distribution, and consider more in depth-broadly the implications of this study for permafrost distribution. Second, I do not see which are the hypotheses the authors want to build. Also for this, I would encourage the authors to either reformulate their

overall aim, or consider the implications of their findings beyond the pure description of the energy fluxes timeseries. With respect to the aim, I am also puzzled by their choice of the model forcing. If the paper's aim is to understand the energy fluxes (and melt and refreezing processes into the soil), then I do not understand why the authors force their model with parameterisations of the radiative fluxes given that they have all measurements available. They instead use the measurements of the four radiative fluxes as a validation of the model, showing indeed that there are differences between observed and modelled shortwave and longwave fluxes. Those differences or errors will translate into errors in the energy fluxes simulated, which are rather gratuitous here. This seems even more important considering that there is no quantification of model uncertainty (see a point below). With the approach they use, they seem to want to test the ability of GeoTOP to parameterise those fluxes. If this is their aim however, this should be stated more clearly, and the paper structured accordingly.

4) Introduction

The Introduction should be substantially improved. The Rationale for this study is not clear and the review of current studies and knowledge gaps is incomplete. There is a single short sentence about precipitation being higher than expected –referring to one single study from 2015 – and then the authors start with "Another key unknown is permafrost. . ." I strongly suggest that the authors provide clear motivations for their work. The overall There is quite a lot of emphasis on permafrost and its potential importance, but the link then to the actual investigation conducted in this paper should be made clearer and stronger. While I overall agree with the authors that "..the knowledge of frozen ground and associated energy regimes are a key knowledge gap in our understanding of the Himalayan cryospheric systems, especially in the Upper Indus Basin", the introduction as it is now does not convey this at all, nor the authors make a compelling case for the motivations for their study.

The aim is general... a foundation for a better understanding of the .. I also question the fact that, being this a point-scale study, the authors cannot say much about the

distribution of permafrost (see my point above).

References and use of literature The authors make extensive use of their own publications to back general statements on the Himalayan cryosphere, but miss the major publications in the field, and especially the many excellent studies from the last couple of years, some of them key papers that have substantially advance our understanding. I find it is not very elegant to refer only to one's own publications, especially when those cannot provide the evidence the authors use them for, as they mostly refer to very local and detailed studies. I would strongly encourage them to use a less parochial approach and give credit to the many excellent studies that have come out recently. The first example is on lines 48-50: "It is hard to propose a uniform framework for the downstream response of these rivers as they originate and flow through various glacio-hydrological regimes of the Himalaya (Thayyen and Gergan, 2010)". That definitely is not the appropriate reference for such a statement, which needs back up from more extensive and comprehensive studies at the scale of the entire Himalaya or HMA and not from one single local catchment in Ladhak. _Argument about permafrost cover being 14 times the one of glaciers should be rephrased, as glaciers have a thickness of hundreds of meters, while permafrost of few meters. I suggest the authors revise those statements. They can still point to the large areas where permafrost is present, but I think they should compare the total amount of ice, e.g. ice volumes or total potential water equivalents and not the area. _ The authors also seem to mix together rock glaciers and permafrost. Are they using rock glaciers as a synonym for permafrost? They should clarify why rock glaciers are mentioned here. There are two theories as regards the genesis of rock glaciers, a glacial and a paraglacial origin, and the authors should make clear that at least they are aware of both.

5) Determination of precipitation The authors use a method called ESOLIP to estimate precipitation from snow depth, which is not described except for the equation used for fresh snow density. I would strongly encourage them to explain at least the basic assumptions of the method in the main text, and include a more detailed description in

the SI, given that snow is an important element of the differences in the two years. The differences between measured and modelled snow depth, listed in Table 1 in the SI, is very high. The authors should justify this.

6) Error estimates in the measurements Both the meteo input and validation datasets lack an assessment of errors. The sensor accuracy is provided in a table but no error estimates are given throughout the paper. They should be included in all figures and tables when comparing observations and simulations.

7) Description of the EB model

This section needs improvement. The sign convention needs to be clarified and improved. It is very confusing. There must be a convention that holds for all fluxes, and then fluxes will be positive or negative based on their direction. This section is rich in some obvious statements, such as that the reflected shortwave radiation is the incoming shortwave radiation times the albedo; and on the other side key information is missing. Here are some of the main aspects/points that should be clearly provided/clarified for the reader to evaluate the model approach and results: _Why do the authors model the longwave radiative fluxes if they are measured? Also, there is no need to list those fluxes' equations, they are very established ones (could be moved to the SI). _On the other side, no info is provided as to the cloud transmissivity, emissivity and other parameters used in those parameterisations, which are really the difficult ones to constrain. _For calculation of the latent heat flux, how is the relative humidity of the surface determined, since it was not measured it seems? _Which are the values of the coefficients alpha and beta in equation 10? The authors should describe what the parameterisations by Pielke et al is based upon, and how the coefficients are calculated, e.g. as a function of which other parameters or variables. In general, values of all model parameters (physical and empirical) should be provided in a Table (see below). _Does the calculation of the turbulent fluxes include corrections for stability of the atmosphere? _How is surface roughness calculated/estimated? The authors should include a table, in the main paper or in the Supplementary Material, where they

include all the values of the soil and surface properties that they use for the model simulations (surface roughness, albedo, conductivity, porosity, etc etc), and an explanation of how those properties were determined. This is important for repeatability but also to understand what modelling choices the authors have made, how sounds they are and how they affect the model output. Most of those properties and parameters are often affected by large uncertainty, which translate into uncertainty in model simulations, so their values should be provided and their uncertainty assessed (see below).

The paper lacks a discussion of the amount of frozen soil that melts and of the corresponding melt water generated by permafrost thawing, which I guess could be calculated with such a model and would be a very useful information to get.

8) Model evaluation: This section is in places redundant, and contains many repetitions. It should be – as the entire paper - reworked and streamlined. For the shortwave radiation: I first of all do not understand why the authors model the shortwave fluxes since they have observations that they can use directly. I think a very strong explanation is needed here if they want to maintain this model forcing. This is important especially because the modelled fluxes do not agree particularly well with the observed ones, see metrics in section 3.5.1 and Figure 5. This is bound to reflect in uncertainties in the simulated energy fluxes. Second, I would disagree with the authors choice of the mean Bias difference and RMSE, and would use instead the NSE, which is more appropriate for variables with a strong temporal cycle, such as runoff, melt rates or indeed shortwave radiation components. The equations of those metrics are not needed, as these are all basic, well known metrics. If they want to include them, I would suggest the authors place them in the SI. In general, I feel that a clear rationale for the use of those many metrics is not clear and should be provided. I do not understand for instance why the authors use distinct sets of metrics for shortwave radiation and ground temperature, which both have a strong sub-diurnal cycle.

9) Partition of fluxes I do not understand how the authors can write that a given amount % of the net radiation was converted into specific percentage of turbulent fluxes: e.g.

"The partitioning of energy balance components during the study period show that 47% of Rn was converted into H, 44% into LE, 1% into G and 7% for melting of seasonal snow", in abstract, line 22-24 and throughout the paper. LE in particular can be both positive and negative, as the authors also show (Table XX). How do the authors quantify percentage fluxes if they have both positive and negative fluxes at any given time? They refer to Zhang et al to calculate the fluxes – but not – I think for the partition of what amount of which flux goes where. They should provide a clear explanation here so that the reader can understand what the values they provide are.

10) Uncertainty analysis One of my main objections to this study is that there is no estimation of uncertainty on the model simulations. I feel that model outputs without an associated uncertainty are no longer acceptable, and I would strongly encourage the authors to do a thorough uncertainty analysis using e.g. a Monte Carlo type of approach, by varying both the meteorological forcing and soil and snow parameters.

11) Figures Figure 3 and 4 should be combined, or presented differently. In its current form, the authors show first the observed radiative fluxes and then the simulated ones – they should be the same or very similar. Indeed, this relates also to one of my objections regarding the forcing of the model: why is the model not forced with the observations of radiative flkuxes, fiven that this is a point-scale application?

The figures showing fluxes over one day, and comparing several days, have little information content. The authors should calculate and plot sub-daily values of fluxes for sub-periods of similar meteorological conditions – if this is their aim – or of similar snow conditions, as one day is really too isolated an example to be significant and representative of a pattern or characteristic.

Figure 8: It is not very informative to present those values for two separate days. I suggest the authors calculate averages for periods of similar conditions.

12) Comparison with other studies This section makes little sense to me. The authors include a comparison also with EB calculations on glaciers, which does not bring, I

feel, many insights to the (very limited) discussion of this paper as glacier surface conditions are very distinct from those that the authors consider at this AWS location. The selections of the sites to include seems arbitrary, and misses numerous EB studies across the world (Wagnon et al., 2009; Pellicciotti et al., 2008; Ayala et al., 2016 Andes; Yang et al 2011, Yang et al 2017, Ding et al 2017, Mölg et al 2012, Mölg et al 2014, Zhang et al 2013 for HMA, and many more for other regions of the world). Also, if this wants to be inclusive: why not including studies of EB and melt regimes over debris covered glaciers, then, which are also abundant (to mention only very few and recent ones: Reid and Brock, 2010, Steiner et al., 2019; Stiglietz et al., 2020) and might be more relevant to permafrost studies than clean ice glaciers? Astonishingly, the authors in their comparison do not consider the elevation of the stations they compare, which plays a key role in determining the amount and sign of fluxes. I would suggest the authors either considerably strengthen this discussion with better argument and a comparison that takes into account at least the differences in elevation, or remove it. Some of the statements provide are obvious and do not add anything to the authors discussion: such as that the albedo of locations with soil or tundra is lower than that of the AWSs on ice (lines 809-811: The mean $\alpha$ for all the sites where radiation balance is measured either on bedrock or tundra vegetation was smaller than those measured over firn or ice during summer"). The authors also do not need to provide those albedo values.

13) Conclusions and main findings This is a mostly descriptive paper, that uses a very complex models but ends up describing mostly the surface energy balance, with very little consideration of the role that permafrost plays in the surface and mass budget. It is very descriptive, and looks more like a report than a scientific paper and I think it would benefit from some more in-depth and perspective. Figures are of poor quality in general, and poorly designed/selected. They often represent times series with little effort of synthesis. There is a long introduction about permafrost and its importance, but the rest of the paper seems disconnected from this focus, and fluxes are not analysed in the context of permafrost characteristics, duration, thawing. The lack of findings and

descriptive nature of this paper is reflected in the fact that indeed the Discussion contains mostly material that should belong to the results. The actual Discussion could definitely be improved.

DETAILED COMMENTS _Line 47: the authors need to provide one ore preferably more references for this statement. _Line 124: what are "strong land-atmosphere interactions"? This is vague and misleading. The authors should reformulate this. _Table 1 Data platform: I guess the authors here refer to the datalogger? _lines 131 to 140: can be removed, or at least substantially shortened or moved to SI. _Line 159-160: remove from there. He authors can put this info in the Acknowledgments if they want. _line 234: strange language, and unclear ("But in Geotop (endrizzi et al., 2014) the equations are described separately"), which should be reformulated. What does it mean and does it bear any relevance for this paper? Do the authors modified some of the formulations in the mode? _Table 4: I would provide the incoming and reflected, incoming and outgoing fluxes separately for the shortwave and longwave radiative fluxes separately. _section 4.1: this entire section belongs to Results. _Lines 695-697: There is no proof here that they are credible. This is a circular argument. _Line 772: (d) high latent heat due to snowmelt that is a heat sink: not clear what the authors man here.  

Please also note the supplement to this comment:
https://tc.copernicus.org/preprints/tc-2019-286/tc-2019-286-RC2-supplement.pdf

---

## Referee Comment (RC3) · Giacomo Bertoldi (Referee) · 9 Jul 2020

Review of the Article Tc-2019-286

The surface energy balance in a cold-arid permafrost environment, Ladakh Himalaya, India

by John Mohd Wani, Renoj J. Thayyen, Chandra Shekhar Prasad Ojha, and Stephan Gruber

The paper presents the results of a monitoring and modeling study to understand the surface energy budget in relation to permafrost formation in a little studied environment, as the high elevation dry Himalayan inner range.

[Figure]

General comments:

This is a valuable and interesting paper, which shows an accurate modelling study of the characteristics of the surface energy budget in a poorly studied region, the inner Himalaya. Moreover, in this region permafrost is widespread and permafrost processes relevant for water resources and risks management. The topic is therefore relevant and the paper suitable for TC. The paper shows new observations and applies with good results a hydrological model, which considers explicit water and energy budget, in a cold and dry catchment.

The modelling study is solid and well done. Model validation convincing. Therefore the methodology appears to be sound (I have only one doubt related to water budget).

However, I have several major comments that, on my opinion, should be addressed before publications, regarding the paper organization and the results discussion. • I suggest to move the model validation section before the discussion of the results. The reader before wants to understand the model′s reliability, and then look to the results on the energy budget. • The presentation of the results is rather long and with many repetitions. The main message of the paper is rather simple. In Ladakh mountain the environment is dry, cold and sunny. Therefore, this leads, compared to other sites, to little incoming longwave and more direct solar radiation which helps permafrost. Snow comes relatively late and major differences are related to the snow duration. This could be explained in a more concise way, leaving space for a more quantitative discussion (see specific comments). • For the methodology, it is not clear to me if soil moisture is explicitly modelled or not (see specific comment at line 210). This has strong implications on the interpretation of the results. • The paper is interesting, but the story is simple. I have the feeling that there are repetitions and details not needed. • I think that the paper could be strongly improved if the model is used also for numerical experiments for quantitatively understand role of climate and possible changes for future permafrost development.

[Figure]

Specific comments: see attached document

Please also note the supplement to this comment:
https://tc.copernicus.org/preprints/tc-2019-286/tc-2019-286-RC3-supplement.pdf

———————————————————————

[Figure]

**Supplement:**

**Review of the Article Tc-2019-286**

**The surface energy balance in a cold-arid permafrost environment, Ladakh Himalaya, India**

**by John Mohd Wani, Renoj J. Thayyen, Chandra Shekhar Prasad Ojha, and Stephan Gruber**

The paper presents the results of a monitoring and modeling study to understand the surface energy budget in relation to permafrost formation in a little studied environment, as the high elevation dry Himalayan inner range.

**General comments:**

This is a valuable and interesting paper, which shows an accurate modelling study of the characteristics of the surface energy budget in a poorly studied region, the inner Himalaya. Moreover, in this region permafrost is widespread and permafrost processes relevant for water resources and risks management. The topic is therefore relevant and the paper suitable for TC.
The paper shows new observations and applies with good results a hydrological model, which considers explicit water and energy budget, in a cold and dry catchment.

The modelling study is solid and well done. Model validation convincing. Therefore the methodology appears to be sound (I have only one doubt related to water budget).

However, I have several major comments that, on my opinion, should be addressed before publications, regarding the paper organization and the results discussion.

- I suggest to move the model validation section before the discussion of the results. The reader before wants to understand the model´s reliability, and then look to the results on the energy budget.
- The presentation of the results is rather long and with many repetitions. The main message of the paper is rather simple. In Ladakh mountain the environment is dry, cold and sunny. Therefore, this leads, compared to other sites, to little incoming longwave and more direct solar radiation which helps permafrost. Snow comes relatively late and major differences are related to the snow duration. This could be explained in a more concise way, leaving space for a more quantitative discussion (see specific comments).
- For the methodology, it is not clear to me if soil moisture is explicitly modelled or not (see specific comment at line 210). This has strong implications on the interpretation of the results.
- The paper is interesting, but the story is simple. I have the feeling that there are repetitions and details not needed.
- I think that the paper could be strongly improved if the model is used also for numerical experiments for quantitatively understand role of climate and possible changes for future permafrost development.

**Specific comments:**

**1. Introduction**

See general comments. More specifically:

**L75** "*The energy balance at the earth's surface drives the Spatio-temporal variability of ground temperature*"

This is an important point, which needs further clarification, since it motivates the rationale of this work. This is mediated by the ground heat flux (both in term of heat diffusion and heat transport by water). A little bit more of basic theory or an equation could help.

**2. Material and methods**

**L 125 – 135: catchment description.** All this information on geology is ok, but at the end what matters are the implications for soil and shallow rock hydraulic and thermal properties. What do you know about them?

**L 210** "*In this study, only the energy fluxes over the snow cover and the ground surface in one-dimensional (1D) mode of GEOtop are used.*" Here is not clear to me if you run GEOtop only in energy budget mode or you are also simulating the soil column water budget. This has strong implications on the interpretation of the results. In the first case, the soil is assumed always saturated and therefore ET from soil could be only potential. In the second case, the soil can become dry and ET is real and can be low in dry snow free periods. Please clarify this important point.

**L 246** - „*Albedo*". It could be interesting for the reader to explain briefly how albedo is changing with respect to snow age and solar angle in GEOtop.

**L 295** - „*Heat equation*". Is GEOtop able to simulate also the heat transport by the water into the soil? This is a very relevant process for permafrost melting (see recent Ph.D. work of Alessandro Cicoria).

**L 305** - „*Snow modelling*". A little bit more details could be useful. At least to say that GEOtop uses a multi-layer, energy based, Eulerian snow modelling approach.

**L 305** - „*performance statistics* ". Okay, but it might be more concise. All is well known.

**3. Results**

I suggest moving the paragraph "Model Evaluation" at the beginning of the results section.

**3.1 Meteorological characteristics**. A lot of details, some of them are not necessary. May be a chart with the difference GST – TA is more informative than many words.

**L 433 - 445 Precipitation**. This section is quite confusing. You have a "measured total precipitation" and then a "precipitation estimated with ESOLIP". It is not clear the difference and the meaning. I guess your measured precipitation is only the liquid precipitation measured by the (unheated?) rain gauge. The ESOLIP precipitation is the sum of the liquid precipitation of the raingauge (with some wind under catch corrections too ?) and of the solid precipitation estimated from snow height data. At the end, later (Figure 6) you find that the ESOLIP precipitation is a more correct estimation. Is this right? Please rephrase this part. If the model evaluation section is before, then the story becomes clearer.

**L 473 Albedo**. This is super low! Over snow covered terrain albedo should be 0.9 – 0.7 minimum, over bare soil around 0.2. Your value is so low because the assumption albedo=0 during the night? During the night albedo is not defined.

**L 500 - 515**. This is also long and boring …

**Figure 4**. Nice Figure. Your story is already there but the reader needs to wait the discussion to figure out what is striking from the Figure. Interesting is the very high sublimation (typical of arid climates – see Herrero works) and the relevant energy absorbed by snow melt (evident in Table 4) in snowy winters which is not going into the soil and therefore is not available for permafrost.

However, I have a question. More snow melt means also more water infiltrating in the soil. How is this water affecting the permafrost?

**3.5 Model evaluation**. Please move this section before. In general, the model performs quite well, and his estimation of the surface fluxes could be considered reliable. Please consider uploading this test case in the testing suite of the GEOtop model website.

**4 Discussion**

**Figure 8:** Choosing two arbitrary days is not very informative. It could be nicer to show the average daily cycle for many snow covered and not snow covered days for the two seasons.

**L 714 - 720** 1% difference seems to be not so significant, given the high uncertainty in surface fluxes estimation. However, the difference from the Figures is quite evident. I do not understand this section.

**L 730 - 745** Ok, the story is clear! Please stop repeating.

**Figure 9 Sub charts E and F.** Why they are informative? I do not understand …

**4.2 Influence of snow cover**. The comparison among two years is interesting, but two years is too less. More years are needed to have general conclusion.

**Line 778** and **Figure 10.** "*Not linear behavior*" Interesting, but the simulated period is too short. You could take advantage from the calibrated model to generate many synthetic years with more and less snow cover. In this way you can generalize the relationship with a numerical experiment … for example increasing or decreasing the precipitation to generate different snow duration and then derive the relation of Figure 10 in a more robust way.

**4.3 Influence of snow cover**. The comparison is interesting, but the characterization of the sites is very different. It seems a part put there having the feeling there is too less in the paper. If you want to make the paper more robust, I suggest performing numerical experiments.

**Minor comments:**

**L 74** – "Spatio" lowercase

**L 205** – GEOtop model references – "*Previous studies have successfully applied GEOtop in mountains regions, e.g., simulating snow depth and ground temperature (Endrizzi et al., 2014), snow cover mapping (Dall'Amico et al.,2018; Dall'Amico et al., 2011; Zanotti et al., 2004), ecohydrological processes (Bertoldi et al.,2010), modelling of processes in complex topography (Fiddes and Gruber, 2012), permafrostdistribution (Fiddes et al., 2015) or modelling ground temperatures (Gubler et al., 2013)*"

Major GEOtop reference, besides Endrizzi et al (2014) is Rigon et al (2006). For ecological processes better cite Della Chiesa et al 2014 or Bertoldi et al 2014. For ground temperatures, besides Gubler et al., 2013, you cold cite Bertoldi et al 2010, which deal on LST modeling in complex terrain. For full reference list please see: https://github.com/geotopmodel/geotop/blob/master/README.rst

**L 220** – "*But in the GEOtop (Endrizzi et al., 2014) the equations of SEB are described separately*" This sentence seems isolated from the context and needs to be revised.

**L 322** – the model was initialized at a uniform **soil** temperature

**References**

*Herrero, J., Polo, M.* (2012). **Parameterization of atmospheric longwave emissivity in a mountainous site for all sky conditions** Hydrology and Earth System Sciences 16(9), 3139-3147. https://dx.doi.org/10.5194/hess-16-3139-2012

*Herrero, J., Polo, M., Moñino, A., Losada, M.* (2009). **An energy balance snowmelt model in a Mediterranean site** Journal of Hydrology 371(1-4), 98-107. https://dx.doi.org/10.1016/j.jhydrol.2009.03.021

*Cicoria A.* **On the dynamics of rock glaciers PhD thesis**, 2020.

---

## Author Comment (AC1) · 10 Oct 2020

**Author response R#1**

 **"The surface energy balance in a cold-arid permafrost environment, Ladakh, Himalaya, India"**

**John Mohd Wani, Renoj J. Thayyen, Chandra Shekhar Prasad Ojha, and Stephan Gruber**
* * *
**Response to Anonymous Referee #1**
* * *
Thank you very much for your review and your constructive comments on this manuscript. I hope that the explanation given below, and the changes to the manuscript, will provide an adequate response.

**General comments:**

| Reviewer comments | Author response |
|---|---|
| - Page 11: When presenting the energy balance equation, the authors use in my opinion a slightly confusing convention related to the flux directions. I would suggest that they use a more common convention very often used in cryospheric research that all fluxes towards the surface are positive and negative away from the surface, because the authors used a different convention often in the paper values are not clearly presented. As an example, in Table 2: the mean value of Sout is given as a negative value and the Min and Max values are given as positive values. The same is the case for LWout. | As suggested, the sign convention for surface energy balance (SEB) components is changed in the revised manuscript. |
| - Page 11: The authors present as their first objective on page 5: (a) quantify the point Surface Energy Balance (SEB)! When calculating the energy balance from measurements, it is then not clear, why the authors do not use their data to calculate the melt by using their measurements of the snow cover? I understand that they use the model to calculate the melt and also the ground temperatures with their model and use the measured data of snow cover and ground temperature as validation data. However, I have the impression that through this approach the authors mix different steps in the methodology and increase the degree of | Combining this suggestion with that of Rev-2 (Comment: 8) we now use the observed radiation components in the GEOtop as input except $LW_{out}$. The comparison of observed and Modelled SEB components is treated separately to assess the model reliability. With this we maintain the two step performance evaluation of GEOtop: 1. modelling and comparison of snow depth variations, and 2. near-surface ground temperature variations and compare with the field observations |

| | |
|---|---|
| freedom unnecessarily. First, the authors may simply use all the available data to determine the MEASURED SEB based on the well-known and common approaches and then in a second step they make their model exercise, which is already very well done. | |
| - Page 22, Table 2: the given albedo values seem to be not reasonable. The authors present for example (taken from Figure 3) measured SWin values in spring (April) of around 300 W m$^{-2}$ and SWout values of 250 Wm$^{-2}$. A corresponding value of albedo (alpha) would be higher than 0.5. Therefore, the max value of alpha should be higher. Please clarify! | We thank the reviewer for pointing out the error in the calculation of albedo and is now corrected in the revised manuscript. The lower values of mean daily albedo in the previous version of the manuscript were due to wrong averaging (used 24 hr.). Now it is corrected. |
| - Table S2 in supp. material: I would recommend that you send your data to the Global Energy Balance Archive (see also http://www.geba.ethz.ch and https://www.earth-syst-sci-data.net/9/601/2017/) | Will do so after getting necessary permission from the funding agency. |

**Specific comments**:

| Reviewer comments | Author response |
|---|---|
| 1. Page 4, line 83: please cite here a text book such as Oke 1987 or Sellers 1965, because these are well-known facts since starting EB studies. | The references suggested have been added to the revised manuscript. |
| 2. Page 10, line 218: what means 'controlled through parameters' -> please be more specific and explain more in detail. | 'Controlled through parameters' here means that the individual processes like surface energy balance or water balance in the GEOtop model can be flexibly controlled separately using the values 1 (on) or 0 (off) in the GEOtop input parameter file. The value equal to 1 means the said process is running and the value 0 means it is turned off. More detail is added in the revised manuscript. |
| 3. Page 10, line 219: please delete s: …mountain regions… | Deleted as suggested in the revised manuscript. |

| 4. Page 11, line 234: replace But with However, | Changed as suggested in the revised manuscript. |
|---|---|
| 5. Page 11, line 240/41: equation (4): why should LWn be only a function of Ts? Please delete the dependencies to Ts in equation, because further down the authors explicitly explain that these variables are not only depending on Ts. | In equation 4, the idea behind showing dependencies was to show that the Eq. 4 is solved in terms of Ts. Yes, the LWn not only depends on Ts through LWout: $$LW_{out} = \epsilon_s \; \sigma T_s^4$$ but also on LWin. In the revised manuscript, the sentences have been reformulated. Furthermore, we have stated that only the LE component in Eq. 4 depends on the soil moisture at the surface ($\theta_w$), which combines the surface energy balance with the water balance equation. |
| 6. Page 12, line 257-260: this is strongly dependent on the effective soil conditions, if you have rock surfaces it is completely different from fine sedimentary material. -> please clarify! Please explain in more detail the BATS, which is used here! | The albedo in GEOtop is considered as per the ground surface conditions such as, for the snow-free ground, the albedo varies linearly with the water content of the topsoil layer, and for snow-covered surfaces the albedo is estimated according to the Biosphere Atmosphere Transfer Scheme. In the GEOtop input parameter file, four parameters need to be defined that take care of soil moisture conditions and their effect on albedo. The values of these parameters were taken from the literature and are described in detail in the revised manuscript. Furthermore, the Biosphere-Atmosphere Transfer Scheme (BATS) (Dickinson et al., 1993), is described in detail in the revised manuscript. |
| 7. Page 14, line 296-298: what happens if your surface is bedrock? | If the soil type is bedrock, then in the input parameter file of the model, the parameters specific to bedrock needs to be defined separately. |
| 8. Page 16, line 360: I would also like to see an evaluation of the turbulent heat fluxes! | The observed values of turbulent fluxes are not available for this study. That's why we did not perform an evaluation of the turbulent heat fluxes. |

| | |
|---|---|
| 9. Page 17, line 387: delete s: root mean square error | Change made in the revised manuscript. |
| 10. Page 20, Figure 2: would be nice to plot snow height in figure 2 A! | The snow height is added to the Figure 2A in the revised manuscript. |
| 11. Page 21, line 468: what do mean with non-free? clarify! | The word is non-snow period and is corrected in the revised manuscript. |
| 12. Page 22, line 476: please reformulate the following sentence to: …with higher values during summertime and low, relatively stable values during winter… | Changed as suggested in the revised manuscript. |
| 13. Page 22, line 481: please reformulate the following sentence to: …with a thick snow cover during… | Changed as suggested in the revised manuscript. |
| 14. Page 22, line 483: please delete word: values | Deleted as suggested in the revised manuscript. |
| 15. Page 23, Table 2: please control and adapt table 2 according to my comments under General Remarks. | In Table 2, the revised albedo values have been updated. |
| 16. Page 27, Table 3 and page 30 line 615: Fsurf values: please explain the signs of these values? Please also explain the variability of Fsurf in relation to your result outputs of your model? What is the meaning of Fsurf when it is negative and there is no snow? Please clarify! | The $F_{surf}$ symbol in the manuscript indicates the latent heat storage in the snowpack due to melting or freezing. During the summertime, when conditions for snow melting are prevailing at the ground surface, the $F_{surf}$ is negative (loss from the system as per revised sign convention) as a result of energy available for melting snow. As per the revised sign convention, the positive $F_{surf}$ (gain to the system) during summertime is the energy used to refreeze the water and represents the freezing flux. Otherwise, the $F_{surf}$ is the soil heat flux for the rest of the time (see Figure 4C). |
| 17. Page 27, line 615: please correct: available | Corrected in the revised manuscript. |
| 18. Page 35, Figure 8: here it is important that most of the energy in Rn is used for melting (particularly in the year 2017) and this should be shown in the figure! | The corresponding snow melt is also shown in the revised figures. |

**References**

Dickinson, R. E., Henderson-Sellers, A. and Kennedy, P. J.: Biosphere-atmosphere transfer scheme (BATS) version 1e as coupled to the NCAR community climate model., 1993.

---

## Author Comment (AC2) · 10 Oct 2020

**Author response R#2**

**"The surface energy balance in a cold-arid permafrost environment, Ladakh Himalaya, India"**

**John Mohd Wani, Renoj J. Thayyen, Chandra Shekhar Prasad Ojha, and Stephan Gruber**
* * *
**Response to Anonymous Referee #2**
* * *
Thank you very much for your review and your constructive comments on this manuscript. I hope that the explanation given below, and the changes to the manuscript, will provide an adequate response.

**Main comments:**

| Reviewer comments | Author response |
|---|---|
| **1. English and style:**
The English contains grammar errors (too many to detail here, but as an example often the third person plural is used when it should have been singular) and weird sentences. The writing style is often redundant and contains many repetitions – I have indicated some below. I had started suggesting corrections to the English but then stopped as this would take many pages and a lot of time. The paper however needs a careful and extensive proof-reading of both English and writing style, and the authors should make an effort to turn the manuscript into a more readable, polished and compelling paper. | The proof-reading of the revised manuscript is done with the help of Grammarly software (Institute Premium License). |
| The abstract seems long and could be shortened and made more to the point. | In the revised manuscript, the abstract is shortened. |
| There are many repetitions in the paper, e.g. lines 242-243; 344-346; 511-512; and in many other instances. | The repetitions in the revised manuscript have been corrected. |
| I would strongly encourage the authors to go through the manuscript and polish/improve it substantially. In its current form, it is not appropriate for publication. | Thanks to the reviewer suggestions, the revised manuscript is presented in a much better way. |
| **2. Paper structure:**
I feel the paper structure needs to be improved in several places. | Thanks to the reviewer suggestions, the revised manuscript is structured in a much |

| | |
|---|---|
| First, I would suggest that the authors separate the study site and data section from the methods section, for readability. As it is now, they need many sub-sections to accommodate all this section content and this section is very long. | better way. The study area and data section are separated from the methods section. |
| Second, a lot of text that belongs to the Methods is contained in the Results section, to a point that the paper is extremely repetitive. Examples are on lines XXX | The repetitive text from the results section is removed. |
| Third, I would encourage the authors to reconsider the way their Results section is structured: first the observations are presented, then the energy fluxes described, and then those are validated with the observations. Before any discussion of the fluxes, they should be validated – otherwise we do not know on which we can have confidence and on which we can have less. | Combining this suggestion with that of reviewer#3 (Comment: 3. Results), the model evaluation section is now moved at the start of the results section in the revised manuscript. |
| I also have some major comments on the figures in this section, which are repetitive and do not make a very good use of space (see comment on Figures below). Finally, most of the content in the Discussion, and most of those figures, should actually be in the Results section, as they present the actual energy fluxes that are the main focus of this paper. | The figures are improved in the revised manuscript. As suggested some part of the discussion is moved to the results section. |
| **3. Aim of the paper:** It is not clear what the paper aim is. The authors state: "we aim to provide a foundation for better understanding the micro-climatological drivers affecting permafrost distribution and temperature regimes in the area, to build hypotheses about similarities and major differences with other, better-investigated permafrost areas". | Aim of the paper is made clear as follows: 1. Understanding the SEB dynamics in a hitherto unknown permafrost area in the UIB. 2. Model seasonal snowpack response (accumulation and melting) and near-surface ground temperature (GST) giving better understanding of snow precipitation (ESILOP) and GST response. 3. Assess the reliability of GEOtop model with minimum input parameters by comparing with observed radiation components. The idea behind the comparison with other permafrost areas is to understand how different micro-climatological drivers such |

| | as incoming shortwave radiation, relative humidity, etc. is comparing with the Ladakh region. |
|---|---|
| First, it does not seem that this study can contribute to understanding the drivers of permafrost distribution, given that it focuses on one single location. If, however the authors think their analysis can contribute to this, they should devote the discussion to examine how their results about energy fluxes at one location can be relevant for permafrost distribution, and consider more in depth-broadly the implications of this study for permafrost distribution. | Permafrost research in this area is in a very nascent stage, and we aim to generate wider acceptability of permafrost in the Ladakh region and provide a basic understanding of SEB processes for the first time.

Furthermore, our aim in this study is not about permafrost distribution. |
| Second, I do not see which are the hypotheses the authors want to build. Also for this, I would encourage the authors to either reformulate their overall aim, or consider the implications of their findings beyond the pure description of the energy fluxes time series. | Based on the comparison, we draw to the conclusion that in this region the, (a) surfaces being overall colder than at a similar location with more relative humidity, (b) Increased amount of incoming shortwave radiation. This will mean that sun-exposed slopes will receive more radiation and shaded ones less (less diffuse radiation) than in comparable areas, and (c) Increased cooling by stronger evaporation in wet places such as meadows. Where there is enough water, you can cool the ground significantly.

With modified objectives and improvement in the discussion, including implications as mentioned above, we addressed the reviewers concern. |
| With respect to the aim, I am also puzzled by their choice of the model forcing. If the paper's aim is to understand the energy fluxes (and melt and refreezing processes into the soil), then I do not understand why the authors force their model with parameterisations of the radiative fluxes given that they have all measurements available. They instead use the measurements of the four radiative fluxes as a validation of the model, showing indeed that there are differences between observed and modelled shortwave and longwave fluxes. Those differences or errors will | Following upon the suggestion, all the observed radiation (incoming and outgoing shortwave radiation, incoming longwave radiation) fluxes except outgoing longwave radiation are now used as input to the model.
GEOtop model does not have the provision to give outgoing longwave radiation as input. It is estimated from modelled ground surface temperature iteratively. |

| | |
|---|---|
| translate into errors in the energy fluxes simulated, which are rather gratuitous here. | |
| This seems even more important considering that there is no quantification of model uncertainty (see a point below). | Uncertainty analysis of the model using PEST tool is undertaken |
| With the approach they use, they seem to want to test the ability of GEOtop to parameterise those fluxes. If this is their aim however, this should be stated more clearly, and the paper structured accordingly. | In addition to our other objectives, such as energy balance modelling, the reliability of GEOtop model as an objective is also added in the introduction. ( Please see the response to comment no. 3, Page 2). |
| **4. Introduction:**
The Introduction should be substantially improved.
The Rationale for this study is not clear and the review of current studies and knowledge gaps is incomplete. There is a single short sentence about precipitation being higher than expected –referring to one single study from 2015 – and then the authors start with "Another key unknown is permafrost…"
I strongly suggest that the authors provide clear motivations for their work. | The motivation behind this study in the introduction section is presented in a much better way, and more references have been added.

Unfortunately, high elevation precipitation data in this region is seldom available. Added couple of references regarding this aspect. |
| The overall There is quite a lot of emphasis on permafrost and its potential importance, but the link then to the actual investigation conducted in this paper should be made clearer and stronger. | The revised manuscript is now structured in such a way that the paper focusses more on energy balance from a permafrost environment. |
| While I overall agree with the authors that "..the knowledge of frozen ground and associated energy regimes are a key knowledge gap in our understanding of the Himalayan cryospheric systems, especially in the Upper Indus Basin", the introduction as it is now does not convey this at all, nor the authors make a compelling case for the motivations for their study. | Permafrost is not considered or appreciated for Hydrological and climate assessment in the Upper Indus regions in India. This paper is a small first step towards appraising the SEB of one site in the permafrost region so that further studies can be triggered to achieve larger goals.
The Introduction section is presented in a much better way, and more references have been added. |
| The aim is general... a foundation for a better understanding of the .. I also question the fact that, being this a point-scale study, the authors cannot say much about the distribution of permafrost (see my point above). | Our aim in this study is not about permafrost distribution, but to understand the energy balance from a permafrost environment in conjuncture with our earlier study (Wani et al., 2020). |

| | |
|---|---|
| References and use of literature
The authors make extensive use of their own publications to back general statements on the Himalayan cryosphere, but miss the major publications in the field, and especially the many excellent studies from the last couple of years, some of them key papers that have substantially advance our understanding. I find it is not very elegant to refer only to one's own publications, especially when those cannot provide the evidence the authors use them for, as they mostly refer to very local and detailed studies. I would strongly encourage them to use a less parochial approach and give credit to the many excellent studies that have come out recently. | As suggested, more references about the Himalayan cryosphere have been added to the revised manuscript. Agree to the fact that there are number of publication on Hydrology of this region (Upper Indus Basin). However, one can notice that the none of these excellent studies mention about permafrost and its role in regional climate and Hydrology. And this is our prime motivation to take up the permafrost studies in the region. (This aspect is added in the introduction of the revised manuscript). |
| The first example is on lines 48-50: "It is hard to propose a uniform framework for the downstream response of these rivers as they originate and flow through various glacio-hydrological regimes of the Himalaya (Thayyen and Gergan, 2010)". That definitely is not the appropriate reference for such a statement, which needs back up from more extensive and comprehensive studies at the scale of the entire Himalaya or HMA and not from one single local catchment in Ladakh. | As suggested, more references have been added in the revised manuscript. |
| _Argument about permafrost cover being 14 times the one of glaciers should be rephrased, as glaciers have a thickness of hundreds of meters, while permafrost of few meters. I suggest the authors revise those statements. They can still point to the large areas where permafrost is present, but I think they should compare the total amount of ice, e.g. ice volumes or total potential water equivalents and not the area. | The statement is intended to give a sense of permafrost cover in the region. Comparison with glacier ice storage and permafrost ice reserve is not intended. Area of permafrost cover/ thaw does matter in terms of high elevation microclimate and disaster potential. Moreover, what is known today is the area. Ice reserve in the permafrost is not known as yet.

These numbers are based on a coarse scale assessment using reanalysis data. Furthermore, in this region, the focusses of researchers have been limited to snow and glaciers. |
| _ The authors also seem to mix together rock glaciers and permafrost. Are they using rock glaciers as a synonym for permafrost? | In the Himalaya, rock glaciers are studied as it is indicative of discontinuous permafrost in the region. Hence we refer to those |

| | |
|---|---|
| They should clarify why rock glaciers are mentioned here. There are two theories as regards the genesis of rock glaciers, a glacial and a paraglacial origin, and the authors should make clear that at least they are aware of both. | studies to give due regard for the past work on this subject. In Wani et al. (2020), we provided a more reliable assessment of permafrost. The rock glacier studies were referred to provide an honest sketch of the progress made in this region. |
| **5. Determination of precipitation:** The authors use a method called ESOLIP to estimate precipitation from snow depth, which is not described except for the equation used for fresh snow density. I would strongly encourage them to explain at least the basic assumptions of the method in the main text, and include a more detailed description in the SI, given that snow is an important element of the differences in the two years. | The ESOLIP method used for precipitation estimation is described in detail in the revised manuscript. All the equations used in the manuscript are added in the supplementary index material. |
| The differences between measured and modelled snow depth, listed in Table 1 in the SI, is very high. The authors should justify this. | In the supplementary material (Table 1), the difference between the measured precipitation and ESOLIP estimated is due to the under-catch of winter snow recorded by the Ordinary Rain Gauge (ORG). |
| **6. Error estimates in the measurements:** Both the meteo input and validation datasets lack an assessment of errors. The sensor accuracy is provided in a table but no error estimates are given throughout the paper. They should be included in all figures and tables when comparing observations and simulations. | In the revised manuscript, the instrument errors are included in the text as well as in the figures. |
| **7. Description of the EB model:** This section needs improvement. The sign convention needs to be clarified and improved. It is very confusing. There must be a convention that holds for all fluxes, and then fluxes will be positive or negative based on their direction. | As suggested, the sign convention for surface energy balance (SEB) components is changed in the revised manuscript. |
| This section is rich in some obvious statements, such as that the reflected shortwave radiation is the incoming shortwave radiation times the albedo; and on the other side key information is missing. Here are some of the main aspects/points that should be clearly provided/clarified for the reader to evaluate the model approach and results: | In the revised manuscript, all the observed radiation (incoming and outgoing shortwave radiation, incoming longwave radiation) fluxes except outgoing longwave radiation are now used as input to the model. GEOtop model does not have the provision to give outgoing longwave radiation as input. It is estimated from modelled ground surface temperature iteratively. |

| | |
|---|---|
| _Why do the authors model the longwave radiative fluxes if they are measured? Also, there is no need to list those fluxes' equations, they are very established ones (could be moved to the SI). | The equations describing the radiative fluxes are moved to the supplementary index material. |
| _On the other side, no info is provided as to the cloud transmissivity, emissivity and other parameters used in those parameterisations, which are really the difficult ones to constrain. | More information about the parameterisations used for estimation of cloud transmissivity, emissivity, etc. is added in the revised manuscript. |
| _For calculation of the latent heat flux, how is the relative humidity of the surface determined, since it was not measured it seems? | Saturated specific humidity at the surface is estimated using GST. |
| _Which are the values of the coefficients alpha and beta in equation 10? The authors should describe what the parameterisations by Pielke et al is based upon, and how the coefficients are calculated, e.g. as a function of which other parameters or variables. In general, values of all model parameters (physical and empirical) should be provided in a Table (see below). | The values of coefficients for soil resistance to evaporation ($\beta_{YP}$ and $\alpha_{YP}$) used in equation 10 are calculated by a function in the source code of GEOtop. More information about the parameterisation of Ye and Pielke (1993) is added in the revised manuscript. The values of all the model parameters is provided in a table in the supplementary index material. |
| _Does the calculation of the turbulent fluxes include corrections for stability of the atmosphere? | Yes, the atmospherical stability in GEOtop is taken care of through a parameter called as "MoninObukhov". Its values can be as follows: If MoninObukhov = 1 stability and instability considered. Similarly, 2 = stability not considered, 3 = instability not considered, and 4 = always neutrality |
| _How is surface roughness calculated/estimated? | In GEOtop, the surface roughness is given to the model as a parameter. In this paper, the value of 0.01m was used based on similar regions, for example in Tibet Plateau (Wang et al., 2018). Furthermore, a threshold is given to change roughness length to snow-covered values in soil area. For the bare soil, the value of the threshold is equal to zero. For snow, the default value of roughness length equal to 0.1 mm was used. |

| | |
|---|---|
| The authors should include a table, in the main paper or in the Supplementary Material, where they include all the values of the soil and surface properties that they use for the model simulations (surface roughness, albedo, conductivity, porosity, etc etc), and an explanation of how those properties were determined. This is important for repeatability but also to understand what modelling choices the authors have made, how sounds they are and how they affect the model output. Most of those properties and parameters are often affected by large uncertainty, which translate into uncertainty in model simulations, so their values should be provided and their uncertainty assessed (see below). | The values of all the model parameters such as atmospheric, soil, snow are provided in a table in the supplementary index material as suggested. Also, an explanation is provided about the determination of parameters. |
| The paper lacks a discussion of the amount of frozen soil that melts and of the corresponding melt water generated by permafrost thawing, which I guess could be calculated with such a model and would be a very useful information to get. | We respectfully disagree. While such a calculation is part of the ultimate motivation for this study, it would be premature at present. This is because the present study is concerned with improving and understanding our ability to predict the spatial differentiation of ground temperature. To calculate runoff, multi-decadal transient model runs would be needed, together with detailed information on the amount and stratigraphic distribution of ground ice. |
| **8. Model evaluation:**
This section is in places redundant, and contains many repetitions. It should be – as the entire paper - reworked and streamlined. For the shortwave radiation: I first of all do not understand why the authors model the shortwave fluxes since they have observations that they can use directly. I think a very strong explanation is needed here if they want to maintain this model forcing. This is important especially because the modelled fluxes do not agree particularly well with the observed ones, see metrics in section 3.5.1 and Figure 5. This is bound to reflect in uncertainties in the simulated energy fluxes. | In the revised manuscript, all the observed radiation fluxes except outgoing longwave radiation are now used as input to the model. See the reply to Comment No 7 & 3 above. |

| | |
|---|---|
| Second, I would disagree with the authors choice of the mean Bias difference and RMSE, and would use instead the NSE, which is more appropriate for variables with a strong temporal cycle, such as runoff, melt rates or indeed shortwave radiation components. | As suggested, the Nash-Sutcliffe Efficiency (NSE) is added for model evaluation in the revised manuscript. |
| The equations of those metrics are not needed, as these are all basic, well known metrics. If they want to include them, I would suggest the authors place them in the SI. | The equations of the evaluation metrics are added in the supplementary index material. |
| In general, I feel that a clear rationale for the use of those many metrics is not clear and should be provided. I do not understand for instance why the authors use distinct sets of metrics for shortwave radiation and ground temperature, which both have a strong sub-diurnal cycle. | The rationale behind the use of different metrics for radiation and ground temperature is because Expressing MBD and RMSD as per cent makes no sense for temperature because the 0 point of the Celsius scale is arbitrary (in contrast to Kelvin). |
| **9. Partition of fluxes:** I do not understand how the authors can write that a given amount % of the net radiation was converted into specific percentage of turbulent fluxes: e.g. "The partitioning of energy balance components during the study period show that 47% of Rn was converted into H, 44% into LE, 1% into G and 7% for melting of seasonal snow", in abstract, line 22-24 and throughout the paper. LE in particular can be both positive and negative, as the authors also show (Table XX). How do the authors quantify percentage fluxes if they have both positive and negative fluxes at any given time? They refer to Zhang et al to calculate the fluxes – but not – I think for the partition of what amount of which flux goes where. They should provide a clear explanation here so that the reader can understand what the values they provide are. | Yes, the method used in Zhang et al. (2013) was used to calculate the proportional contribution of each flux. We thank the reviewer for pointing out the mistake. The correction is made in the revised manuscript. To quantify the percentage of fluxes, we calculated the mean annual average of each of the individual surface energy balance components (LE, H and G) and then divided these individual averages with the mean annual average of net radiation (Rn). For example: Percentage of Rn converted into LE: $$LE/Rn*100$$ The same procedure is adopted by Liu et al. (2019) (Table 1) to calculate the partition ratios. |
| **10. Uncertainty analysis:** One of my main objections to this study is that there is no estimation of uncertainty on the model simulations. I feel that model outputs without an associated uncertainty are no longer acceptable, and I would | Uncertainty analysis of the model using PEST tool is undertaken |

| | |
|---|---|
| strongly encourage the authors to do a thorough uncertainty analysis using e.g. a Monte Carlo type of approach, by varying both the meteorological forcing and soil and snow parameters. | |
| **11. Figures:**
Figure 3 and 4 should be combined, or presented differently. In its current form, the authors show first the observed radiative fluxes and then the simulated ones – they should be the same or very similar. Indeed, this relates also to one of my objections regarding the forcing of the model: why is the model not forced with the observations of radiative fluxes, given that this is a point-scale application? | In the revised manuscript, all the observed radiation (incoming and outgoing shortwave radiation, incoming longwave radiation) fluxes except outgoing longwave radiation are now used as forcings to the model. |
| The figures showing fluxes over one day, and comparing several days, have little information content. The authors should calculate and plot sub-daily values of fluxes for sub-periods of similar meteorological conditions – if this is their aim – or of similar snow conditions, as one day is really too isolated an example to be significant and representative of a pattern or characteristic. Figure 8: It is not very informative to present those values for two separate days. I suggest the authors calculate averages for periods of similar conditions. | During our seasonal analysis, we saw that all the days without cloud cover during the particular sub-season show more or less same patterns in the amplitude of the energy fluxes. That's why we choose to show two arbitrary days instead of an average.
In the revised manuscript, the average seasonal diurnal values of energy fluxes are shown. |
| **12. Comparison with other studies:**
This section makes little sense to me. The authors include a comparison also with EB calculations on glaciers, which does not bring, I feel, many insights to the (very limited) discussion of this paper as glacier surface conditions are very distinct from those that the authors consider at this AWS location. | At line795, we have already mentioned about the lack of studies with data in the manuscript as: "Although aiming to represent differing permafrost environments, this comparison also includes SEB studies on glaciers for lack of other data.**"** |
| The selections of the sites to include seems arbitrary, and misses numerous EB studies across the world (Wagnon et al., 2009; Pellicciotti et al., 2008; Ayala et al., 2016 Andes; Yang et al 2011, Yang et al 2017, Ding et al 2017, Mölg et al 2012, Mölg et al 2014, Zhang et al 2013 for HMA, and many more for other regions of the world). | As suggested, more energy balance studies have been added in the revised manuscript. |

| | |
|---|---|
| Also, if this wants to be inclusive: why not including studies of EB and melt regimes over debris covered glaciers, then, which are also abundant (to mention only very few and recent ones: Reid and Brock, 2010, Steiner et al., 2019; Stiglietz et al., 2020) and might be more relevant to permafrost studies than clean ice glaciers? | As suggested, more recent energy balance studies have been added in the revised manuscript. |
| Astonishingly, the authors in their comparison do not consider the elevation of the stations they compare, which plays a key role in determining the amount and sign of fluxes. | The elevation of stations is already taken into consideration and is available in Table 5. |
| I would suggest the authors either considerably strengthen this discussion with better argument and a comparison that takes into account at least the differences in elevation, or remove it. | The discussion section is now presented in a much better way in the revised manuscript. |
| Some of the statements provide are obvious and do not add anything to the authors discussion: such as that the albedo of locations with soil or tundra is lower than that of the AWSs on ice (lines 809-811: The mean $\alpha$ for all the sites where radiation balance is measured either on bedrock or tundra vegetation was smaller than those measured over firn or ice during summer"). The authors also do not need to provide those albedo values. | The statements mentioned in the comment are removed and are presented in a much better way in the revised manuscript. |
| **13. Conclusions and main findings:** This is a mostly descriptive paper, that uses a very complex models but ends up describing mostly the surface energy balance, with very little consideration of the role that permafrost plays in the surface and mass budget. | In the revised manuscript, more details about the role of permafrost and its influence on the energy balance are provided. |
| It is very descriptive, and looks more like a report than a scientific paper and I think it would benefit from some more in-depth and perspective. Figures are of poor quality in general, and poorly designed/selected. They often represent times series with little effort of synthesis. | Thanks to the reviewer comments, the revised manuscript is restructured and presented in a much better way. |

| There is a long introduction about permafrost and its importance, but the rest of the paper seems disconnected from this focus, and fluxes are not analysed in the context of permafrost characteristics, duration, thawing. | In the revised manuscript, the main focus is given to the energy balance from a permafrost environment. |
|---|---|
| The lack of findings and descriptive nature of this paper is reflected in the fact that indeed the Discussion contains mostly material that should belong to the results. The actual Discussion could definitely be improved. | The material described in the discussion of the earlier version of the manuscript is moved to the results section. The discussion in the revised manuscript is modified and improved. |

**DETAILED COMMENTS**

| Reviewer comments | Author response |
|---|---|
| _Line 47: the authors need to provide one or preferably more references for this statement. | More references have been added to the sentences mentioned in the revised manuscript. |
| _Line 124: what are "strong land-atmosphere interactions"? This is vague and misleading. The authors should reformulate this. | I can't find this statement in the manuscript?? This line is not present in the online version of the manuscript. And we think a much earlier version of the manuscript is sent to the reviewers. |
| _Table 1 Data platform: I guess the authors here refer to the datalogger? | The word Data platform in Table 1 is replaced with the data logger. |
| _lines 131 to 140: can be removed, or at least substantially shortened or moved to SI. | The line numbers between 131 to 140 have been removed in the revised manuscript. |
| _Line 159-160: remove from there. He authors can put this info in the Acknowledgments if they want. | Moved to the acknowledgements. |
| _line 234: strange language, and unclear ("But in Geotop (endrizzi et al., 2014) the equations are described separately"), which should be reformulated. What does it mean and does it bear any relevance for this paper? Do the authors modified some of the formulations in the mode? | In the revised manuscript, the sentence mentioned in the manuscript is reformulated for better clarity. |
| _Table 4: I would provide the incoming and reflected, incoming and outgoing fluxes separately for the shortwave and longwave radiative fluxes separately. | In Table 4, the incoming and outgoing fluxes are given separately for the shortwave and longwave radiations. |

| _section 4.1: this entire section belongs to Results. | Section 4.1 is moved to the results section. |
|---|---|
| _Lines 695-697: There is no proof here that they are credible. This is a circular argument. | This sentence is reformulated in the revised manuscript. |
| _Line 772: (d) high latent heat due to snowmelt that is a heat sink: not clear what the authors man here. | The heat capacities of the mineral or organic soil material, water, and ice, is relatively small by comparison with the quantity of latent heat of fusion.
 For example: To warm 1 g of ice to 1℃ involves the addition of 2.1 J, however, the 334 J $g^{-1}$ of energy must be added to melt it. Therefore, snowmelt is an energy sink because of the latent heat of fusion (Zhang, 2005). |

**References**

Gubler, S., Endrizzi, S., Gruber, S. and Purves, R. S.: Sensitivities and uncertainties of modelled ground temperatures in mountain environments, Geosci. Model Dev., 6(4), 1319–1336, doi:10.5194/gmd-6-1319-2013, 2013.

Liu, X., Xu, J., Yang, S., & Lv, Y. Surface energy partitioning and evaporative fraction in a water-saving irrigated rice field. *Atmosphere*, *10*(2), 51, doi: 10.3390/atmos10020051, 2019.

Wani, J. M., Thayyen, R. J., Gruber, S., Ojha, C. S. P., & Stumm, D.: Single-year thermal regime and inferred permafrost occurrence in the upper Ganglass catchment of the cold-arid Himalaya, Ladakh, India. *Science of the Total Environment*, *703*, 134631, 10.1016/j.scitotenv.2019.134631, 2020.

Wang, C., Zhang, Z., Paloscia, S., Zhang, H., Wu, F., & Wu, Q.: Permafrost Soil Moisture Monitoring Using Multi-Temporal TerraSAR-X Data in Beiluhe of Northern Tibet, China. *Remote Sensing*, *10*(10), 1577, 2018.

Ye, Z. and Pielke, R. A.: Atmospheric Parameterization of Evaporation from Non-Plant-covered Surfaces, J. Appl. Meteorol., 32(7), 1248–1258, doi:10.1175/1520-0450(1993)032<1248:APOEFN>2.0.CO;2, 1993.

Zhang, T.: Influence of the seasonal snow cover on the ground thermal regime: An overview, Rev. Geophys., 43(4), 1–23, doi:10.1029/2004RG000157, 2005.

Zhang, G., Kang, S., Fujita, K., Huintjes, E., Xu, J., Yamazaki, T., Haginoya, S., Wei, Y., Scherer, D., Schneider, C. and Yao, T.: Energy and mass balance of Zhadang glacier surface, central Tibetan Plateau, J. Glaciol., 59(213), 137–148, doi:10.3189/2013JoG12J152, 2013.

---

## Author Comment (AC3) · 10 Oct 2020

**Author response R#3**

**"The surface energy balance in a cold-arid permafrost environment, Ladakh Himalaya, India"**

**John Mohd Wani, Renoj J. Thayyen, Chandra Shekhar Prasad Ojha, and Stephan Gruber**
* * *
**Response to Referee #3: Giacomo Bertoldi**
* * *
Thank you very much for your review and your constructive comments on this manuscript. I hope that the explanation given below, and the changes to the manuscript, will provide an adequate response.

**General comments:**

| Reviewer comments | Author response |
|---|---|
| • I suggest to move the model validation section before the discussion of the results. The reader before wants to understand the model´s reliability, and then look to the results on the energy budget. | As suggested, the model validation section is moved before the discussion section in the revised manuscript. |
| • The presentation of the results is rather long and with many repetitions. The main message of the paper is rather simple. In Ladakh mountain the environment is dry, cold and sunny. Therefore, this leads, compared to other sites, to little incoming longwave and more direct solar radiation which helps permafrost. Snow comes relatively late and major differences are related to the snow duration. This could be explained in a more concise way, leaving space for a more quantitative discussion (see specific comments). | The revised manuscript is rewritten more concisely. The author response to specific reviewer comments depicts the same. |
| • For the methodology, it is not clear to me if soil moisture is explicitly modelled or not (see specific comment at line 210). This has strong implications on the interpretation of the results. | Yes, the soil moisture is modelled using the parameter "*WaterBalance* = 1" in the GEOtop input parameter file. |
| • The paper is interesting, but the story is simple. I have the feeling that there are repetitions and details not needed. | Permafrost research in this area is in a very nascent stage, and we aim to generate wider acceptability of permafrost in the Ladakh region and provide a basic understanding of SEB processes for the first time. To remove the repetitions, the revised manuscript is rewritten more concisely. The |

| | author response to specific reviewer comments depicts the same. |
|---|---|
| • I think that the paper could be strongly improved if the model is used also for numerical experiments for quantitatively understand role of climate and possible changes for future permafrost development. | Presence and implications of permafrost and its thaw in the UIB region, including Ladakh, is not appreciated so far. Our first aim is to provide irrefutable evidence of permafrost and related processes. This paper is a step towards that effort and used only two years of data, which is available.

We highly appreciate the suggestion of the reviewer but feel that it is beyond the scope of this paper. We will certainly attempt this after generating better data and understanding. |

**Specific comments:**

| Reviewer comments | Author response |
|---|---|
| **1. Introduction**
See general comments. More specifically: | Addressed as above. |
| **L75** "*The energy balance at the earth's surface drives the Spatio-temporal variability of ground temperature*"
This is an important point, which needs further clarification, since it motivates the rationale of this work. This is mediated by the ground heat flux (both in term of heat diffusion and heat transport by water). A little bit more of basic theory or an equation could help. | The more theoretical explanation is added in the revised manuscript. |
| **2. Material and methods** | |
| **L 125 – 135: catchment description.** All this information on geology is ok, but at the end what matters are the implications for soil and shallow rock hydraulic and thermal properties. What do you know about them? | The properties (thermal and hydraulic) of soil and rock were not available in our catchment, and we adopted the values of these properties from Gubler et al. (2013). The work of Gubler et al. (2013) and Engel et al. (2017) provides a good starting point for the selection of values for many parameters. |

| | |
|---|---|
| **L 210** "*In this study, only the energy fluxes over the snow cover and the ground surface in one-dimensional (1D) mode of GEOtop are used.*" Here is not clear to me if you run GEOtop only in energy budget mode or you are also simulating the soil column water budget. This has strong implications on the interpretation of the results. In the first case, the soil is assumed always saturated and therefore ET from soil could be only potential. In the second case, the soil can become dry and ET is real and can be low in dry snow free periods. Please clarify this important point. | Yes, we are estimating the energy budget inclusive of simulating the soil column water budget. |
| **L 246** – „*Albedo*". It could be interesting for the reader to explain briefly how albedo is changing with respect to snow age and solar angle in GEOtop. | More theoretical details about the description of albedo in GEOtop such as its change with respect to snow age and solar angle have been added in the revised manuscript. |
| **L 295** – „*Heat equation*". Is GEOtop able to simulate also the heat transport by the water into the soil? This is a very relevant process for permafrost melting (see recent Ph.D. work of Alessandro Cicoria). | The GEOtop does not simulate the heat transport by water into the soil. |
| **L 305** – „*Snow modelling*". A little bit more details could be useful. At least to say that GEOtop uses a multi-layer, energy based, Eulerian snow modelling approach. | More theoretical details about the snow modelling approach used in the GEOtop model have been added in the revised manuscript. |
| **L 305** – „*performance statistics* ". Okay, but it might be more concise. All is well known. | Combining this suggestion with that of Rev-2, the description of performance statistics is written more concisely, and the equations of the evaluation metrics are added in the supplementary index material. |
| **3. Results** | |
| I suggest moving the paragraph "Model Evaluation" at the beginning of the results section. | As suggested, the Model Evaluation section is moved at the beginning of the results section. |
| **3.1 Meteorological characteristics**. | |
| A lot of details, some of them are not necessary. May be a chart with the difference GST – TA is more informative than many words. | This sub-section "Meteorological characteristics" is rewritten more concisely in the revised manuscript. |

| | |
|---|---|
| **L 433 - 445 Precipitation**. This section is quite confusing. You have a "measured total precipitation" and then a "precipitation estimated with ESOLIP". It is not clear the difference and the meaning. I guess your measured precipitation is only the liquid precipitation measured by the (unheated?) rain gauge. The ESOLIP precipitation is the sum of the liquid precipitation of the raingauge (with some wind under catch corrections too ?) and of the solid precipitation estimated from snow height data. At the end, later (Figure 6) you find that the ESOLIP precipitation is a more correct estimation. Is this right? Please rephrase this part. If the model evaluation section is before, then the story becomes clearer. | Observed precipitation is from Ordinary Rain Gauge (ORG). In summer, rainfall is measured directly and in winter snow periods snow w.e. is measured after melting the ORG catch which is certainly underestimated. In winter snow depth is measured using SR50. So yes ESOLIP presented here is liquid precipitation plus SR50. Here, we had the time resolution problem between total measured precipitation (ORG) and other meteorological forcing's including SR50 snow depth (hourly and recorded by automatic weather station). In ESOLIP we considered liquid precipitation on daily basis only. Furthermore, we run the model twice: (a) first model run was made with precipitation data measured in the field, and (b) second model run was made with the ESOLIP estimated precipitation as input. During the evaluation, we find that when using ESOLIP estimated precipitation as input model performance match very well with the snow depletion (Figure 6). |
| **L 473 Albedo**. This is super low! Over snow covered terrain albedo should be 0.9 – 0.7 minimum, over bare soil around 0.2. Your value is so low because the assumption albedo=0 during the night? During the night albedo is not defined. | We thank the reviewer for pointing out the error in albedo, and this is now corrected in the revised manuscript. The lower values of mean daily albedo in the previous version of the manuscript were due to wrong averaging (used 24 hr.). Now it is corrected. |
| **L 500 - 515**. This is also long and boring … | These lines are rewritten more concisely in the revised manuscript. |
| **Figure 4**. Nice Figure. Your story is already there but the reader needs to wait the discussion to figure out what is striking from the Figure. Interesting is the very high sublimation (typical of arid climates – see Herrero works) and the relevant energy absorbed by snow melt (evident in Table 4) in snowy winters which is not going into the soil and therefore is not available for permafrost. However, I have a question. More snow melt means also more water infiltrating in the soil. How is this water affecting the permafrost? | This is certainly an interesting question, and critical for regional hydrology and permafrost response. However, we do not have an answer at this stage as we are working with a limited data set in this paper. With more years of data, we have plans to run the GEOtop model in distributed mode to study the role of infiltrating snowmelt, routing and hydrology. |

| | |
|---|---|
| **3.5 Model evaluation**. | |
| Please move this section before. In general, the model performs quite well, and his estimation of the surface fluxes could be considered reliable. | As suggested, the Model Evaluation section is moved at the beginning of the results section. |
| Please consider uploading this test case in the testing suite of the GEOtop model website. | As and when the review process is complete, a test case will be shared with the developers of the GEOtop model. |
| **4 Discussion** | |
| **Figure 8:** Choosing two arbitrary days is not very informative. It could be nicer to show the average daily cycle for many snow covered and not snow covered days for the two seasons. | During our seasonal analysis, we saw that all the days without cloud cover during the particular sub-season show more or less same patterns in the amplitude of the energy fluxes. That's why we choose to show two arbitrary days instead of an average. In the revised manuscript, the average seasonal diurnal values of energy fluxes are shown. |
| **L 714 - 720** 1% difference seems to be not so significant, given the high uncertainty in surface fluxes estimation. However, the difference from the Figures is quite evident. I do not understand this section. | The idea behind was to give an overview of the partitioning of the surface energy balance and at the same time, its difference during the two contrasting years. |
| **L 730 - 745** Ok, the story is clear! Please stop repeating. | The repetitions have been removed in the revised manuscript. |
| **Figure 9 Sub charts E and F.** Why they are informative? I do not understand … | Figure 9 (Sub-Plots E and F) describe the monthly average variability of turbulent fluxes (H and LE) during low and high snow years. These subplots give a better overview of how the freezing/thawing processes affect the turbulent fluxes and their variability in the seasonally frozen ground and permafrost regions. For example, in early October (Figure 9E and F), the LE began to weaken up to the December for both the years as the seasonally frozen ground began to freeze. Also, during the summer months, the LE starts to increase due to the availability of moisture. Therefore, the seasonal freezing/thawing of the ground affect the LE causing its rapid decrease/increase. |

| | Similar variability is also reported from the seasonally frozen ground and permafrost regions of the Tibetan plateau (Gu et al., 2015; Yao et al., 2011). |
|---|---|
| **4.2 Influence of snow cover**. | |
| The comparison among two years is interesting, but two years is too less. More years are needed to have general conclusion. | Unfortunately, data is limited. Data is being generated, and we will be able to provide more detailed analysis in the coming years. Please see the answer to comment number 5 (page 2 of this response document). |
| **Line 778** and **Figure 10.** "*Not linear behavior*" Interesting, but the simulated period is too short. You could take advantage from the calibrated model to generate many synthetic years with more and less snow cover. In this way you can generalize the relationship with a numerical experiment … for example increasing or decreasing the precipitation to generate different snow duration and then derive the relation of Figure 10 in a more robust way. | This section has been removed from the revised manuscript due to non-availability of data for more years. |
| **4.3 Influence of snow cover**. The comparison is interesting, but the characterization of the sites is very different. It seems a part put there having the feeling there is too less in the paper. If you want to make the paper more robust, I suggest performing numerical experiments. | Please see the reply to the above comment. |

**Minor comments:**

| Reviewer comments | Author response |
|---|---|
| **L 74** – "Spatio" lowercase | Changed as suggested. |
| **L 205** – GEOtop model references – "*Previous studies have successfully applied GEOtop in mountains regions, e.g., simulating snow depth and ground temperature (Endrizzi et al., 2014), snow cover mapping (Dall'Amico et al.,2018; Dall'Amico et al., 2011; Zanotti et al., 2004), ecohydrological processes (Bertoldi et al.,2010), modelling of processes in complex topography (Fiddes and Gruber, 2012), permafrostdistribution (Fiddes et al., 2015) or modelling ground temperatures (Gubler et al., 2013)*"
Major GEOtop reference, besides Endrizzi et al (2014) is Rigon et al (2006). For ecological processes better cite Della Chiesa et al | Added more references that have successfully applied GEOtop as suggested. |

| | |
|---|---|
| 2014 or Bertoldi et al 2014. For ground temperatures, besides Gubler et al., 2013, you could cite Bertoldi et al 2010, which deal on LST modeling in complex terrain. For full reference list please see: https://github.com/geotopmodel/geotop/blob/master/README.rst | |
| **L 220** – "*But in the GEOtop (Endrizzi et al., 2014) the equations of SEB are described separately*" This sentence seems isolated from the context and needs to be revised. | The sentence mentioned is revised in the revised manuscript. |
| **L 322** – the model was initialized at a uniform **soil** temperature | Added the word "soil" in the revised manuscript. |

**References**

Engel, M., Notarnicola, C., Endrizzi, S., & Bertoldi, G.: Snow model sensitivity analysis to understand spatial and temporal snow dynamics in a high- elevation catchment. Hydrological Processes, *31*(23), 4151-4168, doi: 10.1002/hyp.11314, 2017.

Gubler, S., Endrizzi, S., Gruber, S. and Purves, R. S.: Sensitivities and uncertainties of modelled ground temperatures in mountain environments, Geosci. Model Dev., 6(4), 1319‑1336, doi: 10.5194/gmd-6-1319-2013, 2013.

Gu, L., Yao, J., Hu, Z. and Zhao, L.: Comparison of the surface energy budget between regions of seasonally frozen ground and permafrost on the Tibetan Plateau, Atmos. Res., 153, 553–564, doi: 10.1016/j.atmosres.2014.10.012, 2015.

Yao, J., Zhao, L., Gu, L., Qiao, Y. and Jiao, K.: The surface energy budget in the permafrost region of the Tibetan Plateau, Atmos. Res., 102(4), 394‑407, doi: 10.1016/j.atmosres.2011.09.001, 2011.

---

## Author Response (AR2)

**Author response to editor comments:**

**"The surface energy balance in a cold-arid permafrost environment, Ladakh, Himalaya, India"**

**John Mohd Wani, Renoj J. Thayyen, Chandra Shekhar Prasad Ojha, and Stephan Gruber**
* * *
**Response to Editor comments**
* * *
Thank you very much for your review and your constructive editing/comments on this manuscript. I hope that the explanation given below to the specific comments, and the changes to the manuscript, will provide an adequate response.

Please note the line numbers mentioned in the author response refer to the line numbers of the revised draft.

| Editor comments | Author response |
|---|---|
| Line 10-11: unclear sentence: consider revising. What exactly is the meaning of this statement ? | Line 10-11: In the abstract the sentence is revised as suggested. |
| Line 67-70: this is a very strong statement: are you really sure that no previous study about permafrost in Himalaya is available? This I cannot believe...I would suggest to modify the sentence to a less strong wording | Line 67: The statement is removed from the revised manuscript. |
| Line 72-74: without saying how large either the glacier or the permafrost area is, this statement is meaningless (both could be extremely small, so 22 times larger would have no impact). It is also not necessary to quantify this here. I suggest to delete it and/or just mention that permafrost surface areas can be significantly larger than glacier areas. | Line 67-68: The statement is revised as suggested. |
| Line 77-80: as your study focuses on a single site in Ladakh, the repetitions of all these large-scale details is not necessary for your paper, and can be shortened considerably. | Line 71-72: The numbers are removed in the revised manuscript. |
| Line 81: insert new paragraph here. | Line 73: Inserted as suggested. |
| Line 83-88: see comment above: this is not relevant for your study and can be shortened considerably, if not deleted. If you need the elevation range of discontinuous permafrost, then cite it for your study region only. | Line 75-77: The lines are revised as suggested. |
| Line 110: as this expression has not been introduced yet, you should explain it here (in brackets) | Line 101-102: The surface offset is defined as suggested. |

| Line 131: maximum flow (?) | Line 122-124: The values of daily maximum flow are inserted as suggested. |
|---|---|
| Line 132: repetition. | Line 125: Removed as suggested. |
| Line 185-187: unclear sentence: please make your statement clearer | Line 169-171: The sentence is revised for better clarity. |
| Line 192-208: please considering formatting these 4 ESOLIP steps as vertically numbered and itemised list. | Line 176-194: Changes made as suggested. |
| Line 199: psychrometer constant? | Line 183: The word psychrometer constant is replaced with psychrometric constant |
| Line 201, Eq. 1: u_10? | Line 186: u_10 is described as the wind speed measured at 10 m height. |
| Line 206-208: split into two sentences | Line 192-194: Divided into two sentences as suggested. |
| Line 209: "point surface energy balance" | Line 195: The word "point" is removed from the heading as suggested. |
| Line 221-226: multiple repetition (yellow); can be combined with "modelling of ground temperatures in complex topography" | Line 209-210: The citations are combined into one as suggested. |
| Line 232-234: We use the sign convention that energy fluxes towards the surface are positive and fluxes away from the surface are negative (Mölg, 2004). | Line 218-220: Modified as suggested. |
| Line 246: repetition. | Line 232: Deleted as suggested. |
| Line 256-260: trivial and well-known, should be deleted. | Line 242: Deleted as suggested. |
| Line 265: T_s has already been introduced and does not have to be repeated here, but sigma has not been introduced! | Line 244: removed as suggested. |
| Line 271, Eq. 7: beta and alpha are not explained here. They come later, but you have to at least mention it here. | Line 254-256: alpha and beta are defined as suggested. |
| Line 280: unclear what you mean by this phrase: ground surface ? | Line 260: The word is revised to ground surface. |
| Line 318: what do mean by "finer" here: smaller? or do you mean that the discretisation is finer? Then you have to rephrase the whole sentence. as it is now it does not make complete sense | Line 295-296: Yes, we mean that the snow discretisation is finer. The sentence is revised for better clarity. |
| Line 376: include the year here | Line 347: Inserted as suggested. |
| Line 423-427: are all these seasonal details necessary here? Try to focus only on these values, which are important for later analysis. | Line 395-398: These are necessary as later in the manuscript we are discussing the seasonal variability of the SEB components. |
| Line 445, Figure 5D: Is the wind direction really only covering angles between 50 and 240? Are the other directions (0-50 and 240-360) really never happening or is this a shading or averaging effect? | Line 415-417, Figure 5: This is due to the averaging effect. In the revised manuscript, the 5D is removed and the values in the text are referred to the supplementary Figure S5, where hourly values of WD are plotted against the wind speed. |

| | |
|---|---|
| Line: 450-451: I guess these are sums ? so annual total precipitation ? | Line 419-420: Yes, these numbers are sums of annual total precipitation. The word annual is inserted. |
| Line 472-480: if you don't need these values for the following argumentation, it is not needed to cite them all. They are presented in Figure and Table. Of course, if you need them for an argumentation of specific processes or features they should be mentioned explicitly. As it is now, it is just a long list of values, which are already presented in Figures/Tables and should be omitted/shortened. | Line 441-445: These sentences have been revised and the numbers are removed as suggested. |
| Line 482: see comment above? | Line 447-448: The sentence is revised as suggested. |
| Line 483: start new paragraph here. | Line 449: New paragraph started as suggested. |
| Line 488-491: trivial and well-known, not necessary to repeat here. | Line 453: Removed as suggested. |
| Line 494-496: trivial and well-known, not necessary to repeat here. | Line 456: Removed as suggested. |
| Line 522-524: delete | Line 488: Deleted as suggested. |
| Line 556: unclear: seasonal or diurnal variation ? please rephrase | Line 521: its seasonal as well as diurnal variation. The sentence is revised for better clarity. |
| Line 560-580: if you describe here a figure in detail, it must also appear in the main text. If you put it into the supplementary material, additional text should be added there and not in the main text. As these two figures (S6 and S7) are part of the suppl. material, this corresponding text should be placed also in the suppl. material. | Line 522-525: This part is moved to the supplementary material as suggested and only the significant result is mentioned in the main manuscript. |
| Line 605-606: repetition, should be deleted. | Line 549: The sentence is removed for as suggested. |
| Line 619-621: this is quite a lot of repetitions now. I suggest to delete this sentence for better readability | Line 561: The sentence is removed for better readability as suggested. |
| Line 630-631: is this important? I would suggest to delete it to improve the readability. If you add too many (not significant) details, it is difficult for the reader to notice the important details ! | Line 569: Removed as suggested. |
| Line 631: use either sensible heat flux or 'H', but not H flux. | Line 569: the word 'flux' is removed as suggested. |
| Line 635-636: this is a repetition to the previous section and can be deleted here. | Line 572: Removed as suggested. |
| Line 656-658: repetition | Line 589: Removed as suggested. |

| | |
|---|---|
| Line 661-662: this is again a repetition. If the sentence cannot be modified to include new aspects, it should be deleted. | Line 592: Removed as suggested. |
| Line 671-672: This sentence is out of context here. Either, move it to be included in the text further below or add some more info here: for example, what is meant with "Comparatively"? Compared to what or where? (other sites ?) | Line 606-609: The sentence is moved in the text further and more information is added for better clarity. |
| Line 735-737: this is mainly a repetition to the sentence above. Either combine or delete. | Line 670: The sentence has been removed as suggested. |

Furthermore, all other changes suggested in the text have been complied and can be found in the track change file of the manuscript.